# Structural basis for arginine glycosylation of host substrates by bacterial effector proteins

Jun Bae Park[1], Young Hun Kim[2], Youngki Yoo[1], Juyeon Kim[1], Sung-Hoon Jun[1,3], Jin Won Cho[1], Samir El Qaidi[4], Samuel Walpole[5], Serena Monaco[5], Ana A. García-García[6], Miaomiao Wu[4], Michael P. Hays[4], Ramon Hurtado-Guerrero [6,7], Jesus Angulo [5], Philip R. Hardwidge[4], Jeon-Soo Shin[2,8] & Hyun-Soo Cho[1]

The bacterial effector proteins SseK and NleB glycosylate host proteins on arginine residues, leading to reduced NF-κB-dependent responses to infection. *Salmonella* SseK1 and SseK2 are *E. coli* NleB1 orthologs that behave as NleB1-like GTs, although they differ in protein substrate specificity. Here we report that these enzymes are retaining glycosyltransferases composed of a helix-loop-helix (HLH) domain, a lid domain, and a catalytic domain. A conserved HEN motif (His-Glu-Asn) in the active site is important for enzyme catalysis and bacterial virulence. We observe differences between SseK1 and SseK2 in interactions with substrates and identify substrate residues that are critical for enzyme recognition. Long Molecular Dynamics simulations suggest that the HLH domain determines substrate specificity and the lid-domain regulates the opening of the active site. Overall, our data suggest a front-face $S_Ni$ mechanism, explain differences in activities among these effectors, and have implications for future drug development against enteric pathogens.

[1] Department of Systems Biology, College of Life Science and Biotechnology, Yonsei University, 50 Yonsei-ro, Seodaemun-gu, Seoul 03722, Republic of Korea. [2] Department of Microbiology, Yonsei University College of Medicine, 50-1 Yonsei-ro, Seodaemun-gu, Seoul 03722, Republic of Korea. [3] Center for Electron Microscopy Research, Korea Basic Science Institute, Ochang, Chungbuk 28119, Republic of Korea. [4] College of Veterinary Medicine, Kansas State University, Manhattan, KS 66506, USA. [5] School of Pharmacy, University of East Anglia, Norwich Research Park, Norwich NR4 7TJ, UK. [6] BIFI, University of Zaragoza, BIFI-IQFR (CSIC) Joint Unit, Mariano Esquillor s/n, Campus Rio Ebro, Edificio I+D, Zaragoza 50018, Spain. [7] Fundación ARAID, 50018 Zaragoza, Spain. [8] Severance Biomedical Science Institute and Institute for Immunology and Immunological Diseases, Yonsei University College of Medicine, 50-1 Yonsei-ro, Seodaemun-gu, Seoul 03722, Republic of Korea. These authors contributed equally: Jun Bae Park, Young Hun Kim. These authors jointly supervised: Ramon Hurtado-Guerrero, Jesus Angulo, Philip R. Hardwidge, Jeon-Soo Shin, Hyun-Soo Cho. Correspondence and requests for materials should be addressed to R.H-G. (email: rhurtado@bifi.es) or to J.A. (email: j.angulo@uea.ac.uk) or to J.-S.S. (email: jsshin6203@yuhs.ac) or to H.-S.C. (email: hscho8@yonsei.ac.kr)

Protein glycosylation is a post-translational modification implicated in a wide range of cellular/biological processes, including cell development, signaling cascades, and tumorigenesis[1]. Glycosyltransferases (GTs) catalyze the transfer of a sugar moiety to acceptor substrates and are classified according to their folding as GT-A, GT-B, GT-C[2] or GT-D[3]. Most GT-A fold GTs are single domain proteins that contain a Rossmann-like fold though exceptions to this rule exist[4]. GT-A GTs also have a DxD (Asp-x-Asp) motif, which is required to coordinate the divalent cation (cofactor). The donor substrates include sugar-linked nucleotide diphosphates that also interact with the cofactor. Within proteins as acceptor substrates for GTs, the most prevalent glycosylated amino acids are serine and threonine (O-linked glycosylation), and asparagine (N-linked glycosylation).

Another type of glycosylation was recently reported from studies of bacterial virulence proteins[5–7]. Enteropathogenic Escherichia coli (EPEC) and enterohemorrhagic Escherichia coli (EHEC) express numerous effector proteins[8] which are injected into host cells via a type III secretion system (T3SS) to disrupt host cell functions[9]. The NF-κB transcription factor plays a central role in inducing immune responses against microbial pathogens. Some bacterial effectors suppress NF-κB itself or NF-κB-associated factors[10–14]. The T3SS and many effectors are encoded in the locus of enterocyte effacement (LEE)[15]. Effectors encoded outside this region are designed as non-LEE effectors (Nles)[16]. The non-LEE encoded effector protein B (NleB) has GT activity and inhibits NF-κB activation by transferring N-acetyl glucosamine (GlcNAc) to host death domain-containing proteins and to glyceraldehyde 3-phosphate dehydrogenase (GAPDH)[5–7]. The glycosylation target is an arginine residue, which was unexpected because the guanidine group of arginine is nucleophilically poor at physiological pH.

NleB target proteins include the tumor necrosis factor receptor type 1-associated death domain (TRADD), Fas-associated death domain (FADD), receptor-interacting serine/threonine-protein kinase 1 death domain (RIP1-DD), tumor necrosis factor receptor death domain (TNFR-DD), and GAPDH[5–7]. Most of these proteins participate in regulating the tumor necrosis factor-alpha (TNF-α) mediated apoptosis pathway via death domain mediated homo- or hetero-oligomerization[10]. Previous studies have reported that glycosylation of FADD Arg117 and TRADD Arg235 disrupts apoptosis and decreases NF-κB signaling in host cells[5,6]. Glycosylation of GAPDH Arg197 and Arg200 inhibits ubiquitination of the TNF receptor-associated factors (TRAF) 2 and 3, leading to reduced NF-κB signaling and type I interferon production[17,18].

The T3SS effectors SseK1 and SseK2 from Salmonella typhimurium SL1344 are NleB orthologs that behave as NleB1-like GTs, although they differ in protein substrate specificity[6,18]. The third member, SseK3, is inactive against FADD and GAPDH but active against TRADD[18,19]. Recently, the structure of SseK3 was determined, revealing a GT-A fold[19]. However, the specific enzyme mechanism and the identification of the catalytic base remain unclear. There are also discrepancies regarding whether these enzymes are retaining or inverting GTs because this has not been experimentally probed[19,20]. In addition, details regarding substrate specificity based on structural evidence are also limited due to the lack of ternary complexes. Here, by using a combination of X-ray crystallography, STD-NMR, enzyme kinetics, molecular dynamics simulations, and in vivo experiments, we show that these enzymes are GT-A fold retaining GTs that most likely follow an $S_Ni$ mechanism. We demonstrate that the HLH domain is relevant to protein substrate recognition and the HEN residues are critical for catalysis. We also determine differences within the SseK/NleB family on recognition of the sugar

nucleotide and peptide substrates and find common features for the three peptides such as the recognition of the conserved Trp and Arg residues (WR-motif). Finally, molecular dynamics simulations reveal that the presence of GlcNAc in the donor site induces conformational changes on the side chains of the peptide substrate so that the final arginine acceptor becomes properly oriented for a front face attack to the anomeric $C_1$ carbon of the sugar.

## Results

**Anomeric configuration of glycosylated peptides.** Recently it was proposed, though not experimentally demonstrated, that SseK3 is a retaining GT[19]. However, and in contrast to this, another NleB study synthesized Arg-N-GlcNAc-containing glycopeptides in a β-configuration, implying that these enzymes are inverting GTs[20]. To resolve these discrepancies, we investigated by NMR spectroscopy the glycosidic bond configuration of a GlcNAc-GAPDH$_{187-203}$ glycopeptide, which was formed enzymatically by incubation with SseK1 and UDP-GlcNAc/MnCl$_2$. From 2D $^1H,^{13}C$-CLIP-HSQC we measured the $^1J_{CH}$ coupling at the anomeric position of the transferred GlcNAc to be 168 Hz, characteristic of an α-linkage (Fig. 1). These data suggest that the transfer of GlcNAc by SseK1 follows a retaining mechanism. Considering the conserved active site residues and the structural similarity between SseK1 and SseK2/SseK3, SseK2/SseK3 also might be retaining GTs (detailed information is described below).

**Overall enzyme architecture.** We solved the crystal structures of Salmonella enterica serovar Typhimurium SL1344 SseK1 in complex with UDP. SseK2 was solved both in its unliganded form and in complex with UDP and UDP-GlcNAc. NleB2 from E. coli O145:H28 was solved in its unliganded form. (Supplementary Table 1). For overexpression and crystallization, amino acids 1–20 and 1–33 at the N-terminus of SseK1 and SseK2 were truncated, respectively. This N-terminal region is predicted to be unstructured and presumably play a role in secretion and translocation into the host cell[21]. Proteins containing these N-termini failed to crystallize. Point mutations (C39S, C210S) were introduced into SseK1 to prevent protein precipitation due to irregular intermolecular disulfide binding. For NleB2, amino acids 317–326 at the C-terminus were deleted for better crystal packing and Cys21 and Cys199 were also substituted to serine (Fig. 2a). Unambiguous electron density maps for uridine 5′-diphospho-N-acetylglucosamine (UDP-GlcNAc) or uridine 5′-diphosphate (UDP) were visualized in the active sites of the crystal structures (Fig. 2b).

The sequence identity among SseK1, SseK2, and NleB1 ranges from ~ 60-65%. The N-termini, whose function is presumably to facilitate protein translocation into host cells, differ the most among orthologs, while the rest of the sequence is highly conserved[22]. Hereafter, we will focus primarily on the biochemistry for SseK1 because SseK1 is more active than SseK2. For structural analyses, we focus on SseK2 because the data-sets for this protein were obtained at a higher resolution, with three different snapshots of the active site. SseK2 possesses an overall protein fold composed of 15 α-helices and 9 β-strands that is highly similar to SseK1 and NleB2 (RMSD = 1.8 Å/1.8 Å, Z-score = 37.2/37.4, number of compared residues = 304/304 to SseK1/NleB2, respectively based on DALI pairwise comparison[23]) (Fig. 2c, d). These structures belong to the GT-A class, which has two abutting β/α/β Rossmann-like domains[4] (β3-α2-β4-α3-β5) and contains an Asp-x-Asp (DxD) motif in the active site (SseK2$^{D239-x-D241}$, SseK1$^{D223-x-D225}$, and NleB2$^{D218-x-D220}$).

SseK2 can be divided into three types of sub-domains, namely the catalytic domain (40–147 and 185–336), which includes the

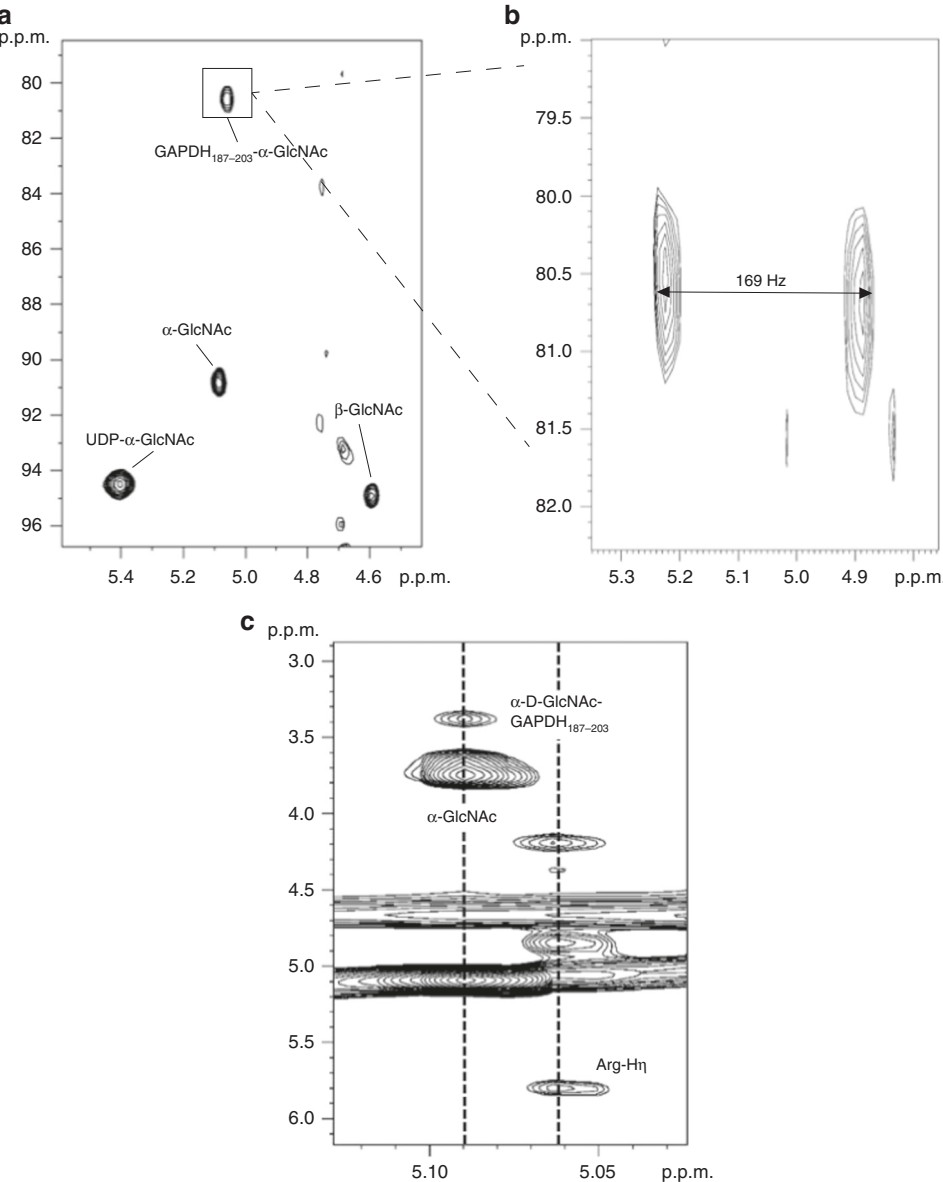

**Fig. 1** SseK1 is a retaining-glycosyltransferase. NMR spectra showing the reaction product of GAPDH$_{195-203}$ with SseK1. **a** Decoupled $^1$H-$^{13}$C HSQC spectrum (800 MHz) showing the anomeric region, highlighting the presence of α-D-GlcNAc-GAPDH$_{187-203}$, with a large $^{13}$C upfield shift relative to the free species. **b** Expansion of $^1$H-$^{13}$C CLIP HSQC spectrum (500 MHz) with no decoupling to measure the anomeric $^1J_{CH}$ coupling in α-GlcNAc-GAPDH$_{187-203}$. A value of 169 Hz indicates an α-configuration. **c** $^1$H-$^1$H TOCSY spectrum highlighting through-bond correlation between the anomeric proton of α-D-GlcNAc in GAPDH$_{187-203}$-α-GlcNAc and an arginine η-proton

Rossmann-like domains, the protruded helix-loop-helix (HLH) domain (148–184), and the C-terminal lid domain (337–348) (Fig. 2e). The concave shape of the catalytic domain is composed of an α-helix and β-strand mixture, and similar to other GT-A structures, continuous central β-strands (β8, β9, β6, β3, β4, β5) form a mixture of parallel and anti-parallel strands. The C-terminal lid domain is highly flexible in the absence of a ligand. Therefore, the electron density map for this domain was not resolved. However, in the structures with UDP and UDP-GlcNAc, the substrate leads to an unambiguous electron density map for the C-terminal lid domain, implying that the domain is well ordered only in the presence of the nucleotide (detailed information is described below).

Recently, the crystal structure of EarP, an arginine rhamnosyl-transferase, was solved, revealing a GT-B fold and an inverting catalytic mechanism in which a glutamate residue acts as the

catalytic base[24–26]. Hence, SseK and EarP are likely to differ in their catalytic mechanisms (see below).

**Donor substrate binding mode.** Based on the complexes of SseK2 with UDP and UDP-GlcNAc, we identified the donor-substrate binding mode and substrate-mediated conformational changes. UDP-GlcNAc consists of three groups, namely the uridine, pyrophosphate, and GlcNAc, which will be discussed independently. The uridine group has an aromatic ring tethered by Phe203 and Trp65 through π-π stacking, water-mediated indirect hydrogen bonds (backbones of Arg68 and Ser346), and hydrogen bonding with the Phe66 backbone (Fig. 3a, upper panel). This sandwich-like π-π stacking is an unusual interaction in GTs because in most of them the sugar nucleotide uracil moiety is sandwiched between an aromatic and an apolar non-aromatic residue participating in π-π stacking and CH-π

 3

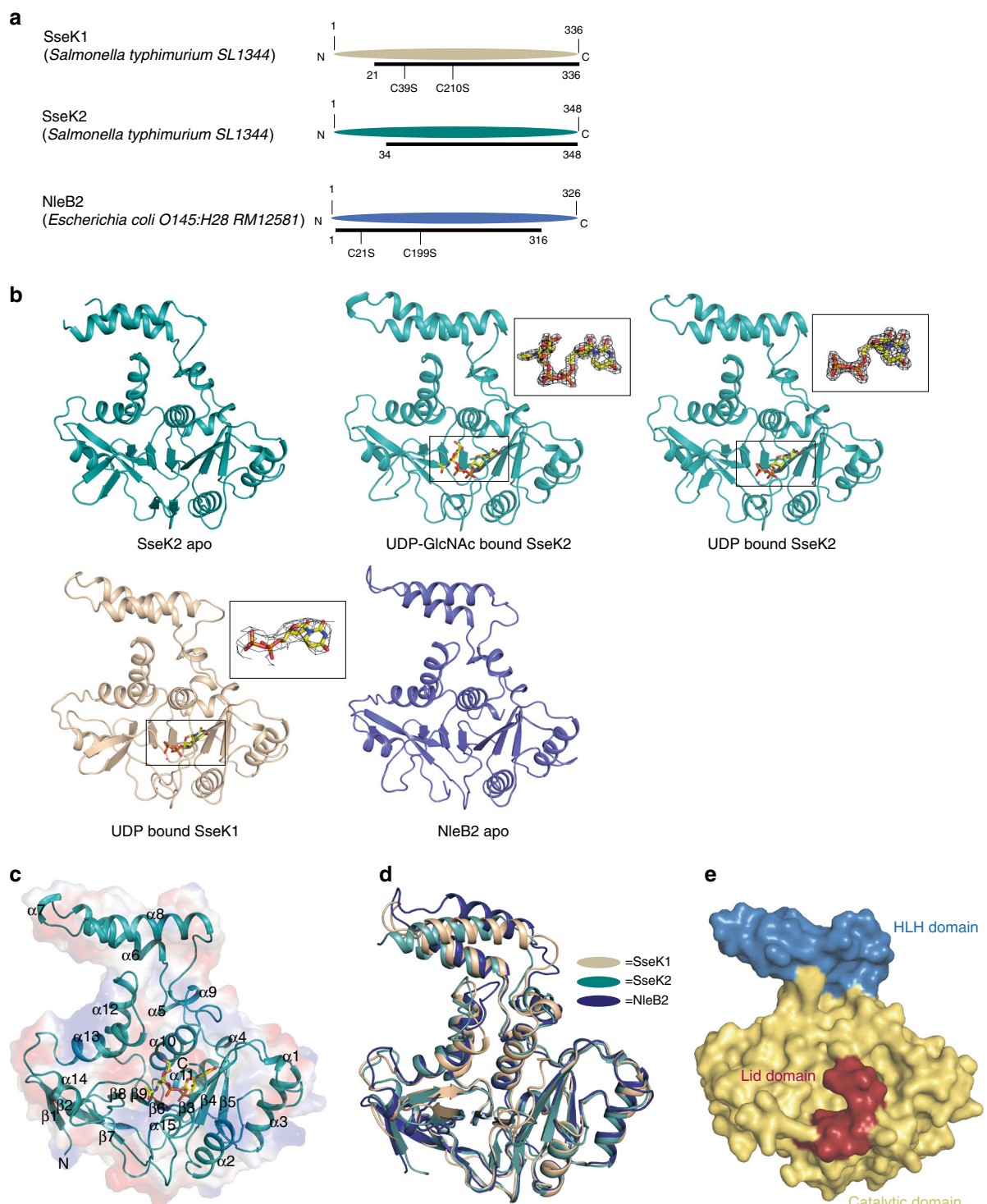

**Fig. 2** Overall enzyme architecture. **a** Colored ovals show the full-length amino acid sequence and the black line under the oval shows the amino acid sequence used for overexpression and crystallization. **b** Arrange crystal structure and electron density maps of each ligands (2Fo – Fc electron density maps of UDP on SseK1 and UDP & L-arginine on SseK2 contoured at 1σ and UDP and the others are 2σ). **c** Numbering of the α-helices and β-strands of SseK2 and **d** superimposition of SseK1, SseK2, and NleB2. **e** Each sub-domain is presented in different colors (blue: HLH domain, yellow: catalytic domain, red: lid domain)

interactions, respectively[27,28] (Supplementary Fig. 1a), implying that this unusual interaction is not a requirement for GTs that prefer uracil-containing sugar nucleotides. However, this sandwich-like π-π stacking interaction is unique for this family of enzymes and is determinant for recognition of the uracil moiety (see below).

Both SseK1 and SseK2 share sandwich-like aromatic π-π stacking interactions and both the tryptophan and phenylalanine residues are conserved in NleB2. However, in the sandwich-like π-π stacking, the interaction modes of SseK1 and SseK2 are slightly different. In contrast to SseK2, Trp331 from the C-terminal lid of SseK1 interacts with the uracil base instead of

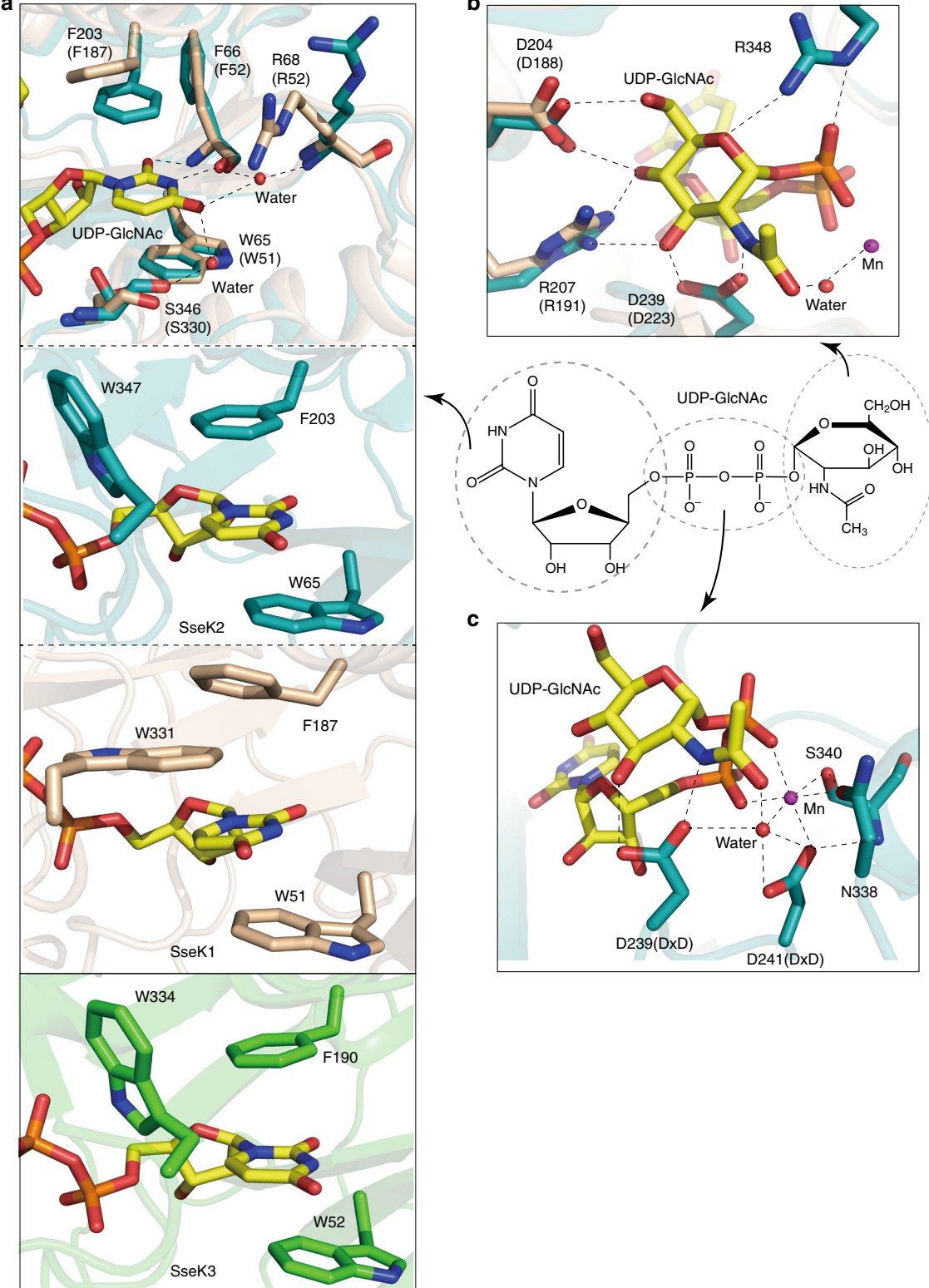

**Fig. 3** UDP-GlcNAc binding mode in SseK2. **a** Uracil moiety of UDP-GlcNAc interacts with SseK2 through hydrogen bonds and π-π stacking (top panel), but SseK1 uses a slightly different mechanism (second and third panel). Uracil binding mode of SseK3 is similar to SseK2 instead of SseK1 (bottom panel). **b** GlcNAc moiety of UDP-GlcNAc interacts with Asp204, Arg207, Asp239, and Arg348 by hydrogen bonds. The carbonyl group of the acetyl of GlcNAc interacts with a water molecule to stabilize the divalent metal ion. **c** Manganese ion coordinates six oxygens from pyrophosphate, Ser340, Asn338, Asp241, and water. The DxD motif stabilizes both UDP-GlcNAc and manganese ion. *Amino acid numbering in brackets refers to conserved sequence of SseK1. Black dashed lines represent hydrogen bonds

Phe203 (Fig. 3a, second and third panels). To confirm this structural difference, we mutated the Phe residue of SseK1 and SseK2 and measured the $K_d$ for UDP-GlcNAc by using isothermal titration calorimetry (ITC) (Supplementary Fig. 1b, c). While the $K_d$ measured for binding of UDP-GlcNAc to SseK1 F187A was similar to wild type SseK1, the $K_d$ for binding of UDP-GlcNAc to SseK2 F203A was increased about 13-fold as compared to wild type SseK2. These data suggest that although the sandwich-like π-π stacking interactions are maintained, the uracil binding modes within SseK1 and SseK2 are slightly different. In the SseK3 structure, SseK2-like π-π stacking interaction is conserved and Phe190[SseK3] and Trp52[SseK3] (corresponding to Phe203[SseK2] and Trp65[SseK2]) participate in an interaction with the uridine group in the same orientation[19] (Fig. 3a, bottom panel). Mutating Trp51[SseK1] and Trp65[SseK2] to alanine abrogated UDP-GlcNAc binding to SseK1 and SseK2 (Supplementary Fig. 1b, c). The activity of the W51A mutant was reduced more than the W331A mutant, as measured in NF-κB activation assays, which was consistent with ITC assay data (Supplementary Fig. 1d). Overall, Trp51[SseK1] and Trp65[SseK2] appear to be more critical than Trp331[SseK1] and Phe203[SseK2] for π-π stacking interactions with UDP-GlcNAc, Note that for all sugar nucleotides the binding energy is dominated by a large negative enthalpic term and to a lesser extent by a non-favored entropic term (Supplementary Fig. 1b, c).

The GlcNAc moiety of UDP-GlcNAc establishes hydrogen bond interactions with Asp204, Arg207, Asp239, and Arg348. The acetyl group of GlcNAc stabilizes the manganese ion by water-mediated hydrogen bonds (Fig. 3b). The importance of the acetyl moiety was confirmed by ITC data that show increased $K_d$ of UDP-glucose (UDP-Glc) and UDP-galactose (UDP-Gal) as compared to the $K_d$ of UDP-GlcNAc for SseK1 (Supplementary Fig. 1b, c). In comparison to UDP-GlcNAc, UDP-Glc lacks the acetyl group, leading to a decrease in the enzyme-substrate binding affinity of about 16.5-fold. Moreover, in UDP-Gal, the absence of the acetyl group and the presence of an inverted $C_4$ hydroxyl group may lead to steric hindrance with the enzyme, leading to the weakest $K_d$ (69.1-fold weaker than the $K_d$ of UDP-GlcNAc). The binding affinities of SseK1 and SseK3 for UDP-Glc and UDP-Gal are relatively different[19]; however, their binding affinities for UDP-GlcNAc are similar (2.3, 1.2, and 1.9 μM for SseK1, SseK2, and SseK3, respectively)[19]. Overall, the SseK enzymes possess an architecture that is optimized for binding UDP-GlcNAc.

Most GT-A GTs have a DxD motif that is required for enzymatic activity[29]. The DxD motif in SseK2 has two significant functions, the coordination of manganese ion and the interaction with the GlcNAc group (Fig. 3c). The manganese ion acts as a bridge between SseK2 and the pyrophosphate of UDP-GlcNAc. In the absence of manganese, the DxD motif-mediated donor-substrate binding would not be expected to occur due to the negative charge repulsion between DxD and the pyrophosphate of UDP-GlcNAc. An octahedral molecular geometry was visualized for the manganese ion coordinated to six oxygens from the UDP pyrophosphate, Asp241, Ser340, Asn338, and a water molecule. Asp239 interacts with a water molecule and a GlcNAc moiety via hydrogen bonds. Asp241 interacts with both a manganese ion and with a water molecule that stabilizes the manganese ion. Most of the residues that interact with UDP-GlcNAc are highly conserved in the SseK and NleB families (Supplementary Fig. 2).

**Conformational change by the donor-substrate binding.** After donor-substrate binding, several GTs undergo large local conformational changes. For example, human glycogenin1 (hGYG1)

has a lid, an acceptor arm, and a C-terminal loop. Their conformational rearrangement influences the accessibility of the substrate at the active site and in turn their catalytic activity[30]. The SseK2 structure undergoes a dramatic conformational change induced by donor-substrate binding. In the ground state of SseK2, the C-terminal lid domain is highly flexible, impeding its visualization in the crystal structure. In this state, the donor-substrate binding site might be fully exposed to allow the access of UDP-GlcNAc (Fig. 4a, left panel). After UDP-GlcNAc binds to the active site, the α10 helix is tilted towards the UDP-GlcNAc by ~ 3.5° and the C-terminal lid domain covers up the active site to stabilize the bound substrate and to restrict the accessibility of water molecules (Fig. 4a, right panel and Fig. 4b). The closure of the C-terminal lid in the presence of UDP-GlcNAc determines a closed conformation for this family of enzymes. We truncated the lid domain (SseK1 1-321) and found that the $K_d$ of UDP-GlcNAc binding increased about 155.3-fold for the lid domain truncation, as compared to the wild type protein (Supplementary Fig. 1b, c), suggesting that the C-terminal lid domain plays a key role for donor-substrate binding.

Both Trp334 and Arg335 in the C-terminal lid domain of SseK3 (corresponding to Trp347[SseK2] and Arg348[SseK2]) interact with UDP-GlcNAc[19]. This interaction is highly similar to that of SseK2. It was reported that Trp334 and Arg335 in SseK3 are essential for enzyme activity[19]. The amino acid sequence of the C-terminal lid domain of SseK1 is slightly different from SseK2 and SseK3. The conserved arginine residue in SseK2 and SseK3 (Arg348[SseK2], Arg335[SseK3]), which is located in the lid domain, is substituted to an alanine residue in SseK1 (Ala332[SseK1]). In addition, an arginine residue is located next to an alanine residue (Arg333[SseK1]). In the crystal structure of UDP-bound SseK1, the backbone direction of Ala332 is located at the opposite side of UDP (Fig. 4c). This implies that the lid domain of SseK1 is likely more flexible than SseK2 and SseK3. Furthermore, the sequence alignment shows that the NleB family lacks the arginine residue, though a conserved Trp is present at the C-terminus. We would expect that both the SseK and NleB families have a different conformational behavior of the lid-domain.

**Peptide substrate recognition by SseK1 and SseK2.** To obtain structural information on the molecular recognition of the substrates, we performed saturation transfer difference (STD) NMR experiments using short peptides from FADD, TRADD, and GAPDH. Standard homo- and heteronuclear 2D NMR techniques were used to obtain the chemical shift assignments of GAPDH$_{195-203}$, FADD$_{110-118}$, and TRADD$_{229-237}$ (Supplementary Table 5–7). For each peptide, four different enzyme systems were prepared: apo-SseK1, apo-SseK2, holo-SseK1, and holo-SseK2, where apo and holo stand for the enzyme without and with Mn$^{2+}$ and UDP, respectively. We observed that all three peptides bound to both SseK1 and SseK2, irrespectively of the forms used in the experiments (Supplementary Fig. 3, 4). These data imply that binding of the short peptide ligands occurs independently of enzymatic activity and can also take place in the absence of the sugar nucleotide. In STD NMR experiments, strong signal intensities from different hydrogen atoms of the ligand permit identification of the main contacts of the peptides with the enzyme in the bound state[31,32]. After intensity normalization, binding epitope maps of the peptides were obtained (Fig. 5 and Supplementary Fig. 5). In all cases, high STD signals, indicating close contacts, were observed for the conserved Trp and Arg side chains. The results support the concept that a WR-motif (W112/R113 in FADD, W230/R231 in TRADD, and W196/R197 in GAPDH) appears to be central for recognition. For TRADD$_{229-237}$, although rather similar binding modes to SseK1 and SseK2

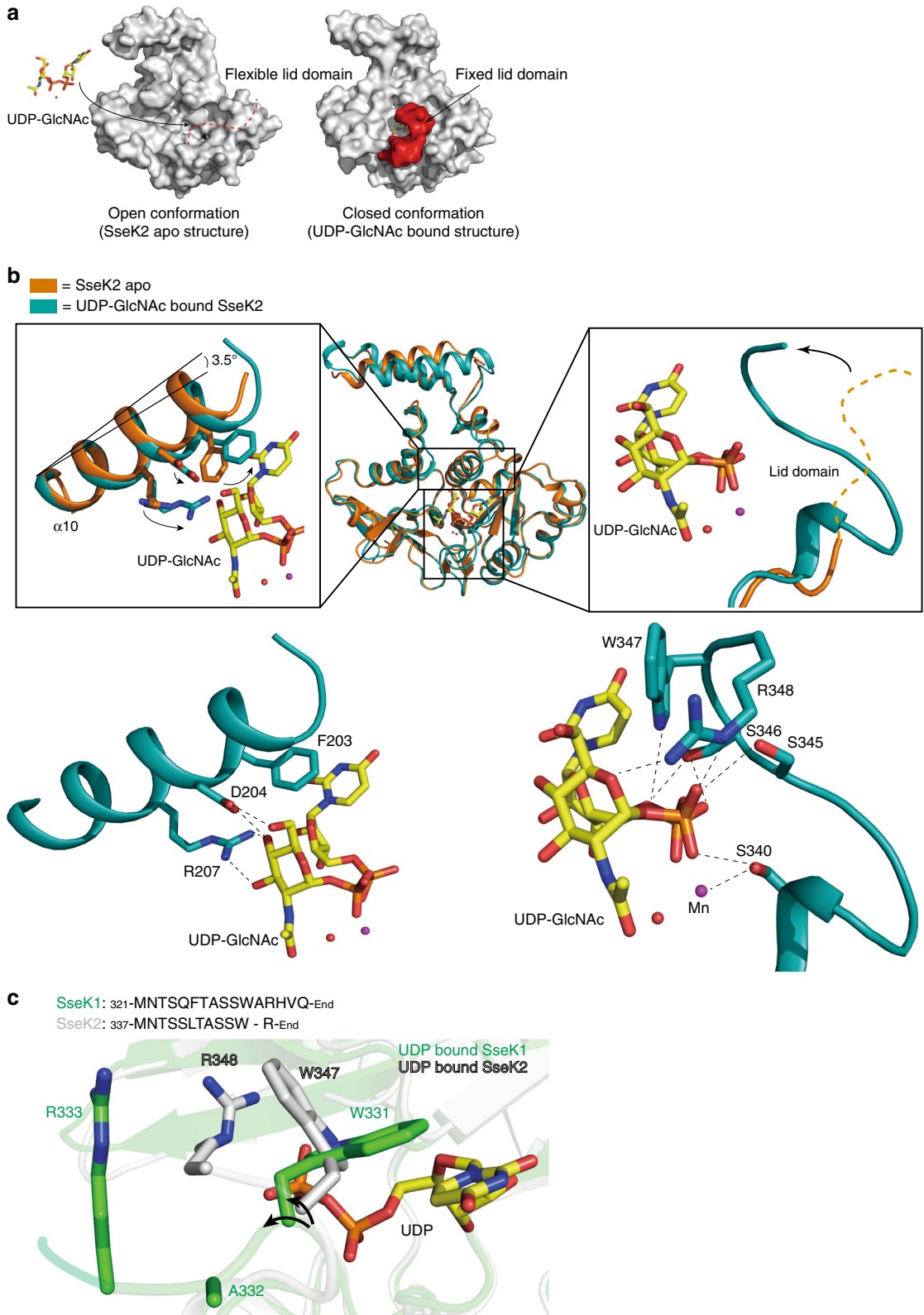

were observed, the binding epitope was spread across the entire molecule when bound to SseK1 but was more concentrated around the WR-motif for SseK2. The epitopes of $FADD_{110-118}$ are comparable when bound to either SseK1 or SseK2. For $GAPDH_{195-203}$, differences in binding to both enzymes were observed. For the binding of $GAPDH_{195-203}$ to SseK1, the data support a significant conformational rearrangement of the peptide ligand upon addition of $Mn^{2+}$ and UDP. This was evidenced by a significant change in the binding epitope mapping, particularly a large increase in STD intensities for the arginine side-

**Fig. 4** Donor substrate-mediated conformational changes. **a** Comparison of surface structures between apo form and UDP-GlcNAc bound SseK2. The flexible C-terminus creates an open conformation for access of UDP-GlcNAc and after binding of the UDP-GlcNAc, the C-terminal lid domain (red color) interacts with UDP-GlcNAc and is fixed in closed conformation. Yellow stick represents UDP-GlcNAc. **b** After the donor-substrate binds to the active site, the α10 helix tilts about 3.5 degrees and the flexible C-terminal lid domain is fixed towards the UDP-GlcNAc. Due to hydrogen bonding at Arg207, Asp204, and aromatic stacking at Phe203, the α10 helix is able to tilt. The flexible C-terminal lid domain can be fixed by hydrogen bonding of Ser340, Ser345, Ser346, Arg348, and Trp347. Orange and blue-green colors represent SseK2 apo and UDP-GlcNAc bound SseK2 structure, respectively. Black dashed lines represent hydrogen bonds. **c** Direction of the lid-domain is different between the SseK1 (in green) and SseK2 (in white) structures. Curved arrows (in black) each correspond to the direction of the backbone of the lid-domain

chains (Fig. 5c, d). However, for all other peptides, and for GAPDH$_{195-203}$ binding to SseK2, no such rearrangement occurs (Supplementary Fig. 6).

To elucidate the specific role of the WR-motif, we measured the kinetics of SseK1 to four GAPDH$_{187-203}$-derived synthetic peptides (designated as WT, W196A, R197A, and W196A/R197A). Each of the W196A, R197A, and W196A/R197A mutant forms decreased the catalytic efficiency of about 40.5%, 47.4%, and 17.3%, respectively as compared to WT GAPDH$_{187-203}$ peptide (Supplementary Fig. 16b). In particular, the double mutant form (W196A/R197A) synergistically decreased enzyme catalysis, supporting our STD-NMR experiments that suggested that the WR-motif of the peptides is of utmost importance for binding to these enzymes.

To investigate the relevance of the WR-motif for binding, we also carried out STD NMR experiments focused on analyzing the impact of single and double mutations on the affinity of the molecular recognition of the synthetic peptide TRADD$_{229-237}$. As we were interested only in analyzing the impact on binding, we ran competition experiments for the interactions of the TRADD$_{229-237}$-derived peptides with SseK1. Five synthetic peptides (designated as WT, W230A, R231A, W230A/R231A, and R235A) were analyzed. All of them bound to SseK1, as detected by STD NMR, yet their affinities were different, as reflected in their differences in average STD NMR intensities (e.g., the most intense alpha proton showed 14%, 9%, 6%, 4%, and 10% for WT, W230A, R231A, W230A/R231A, R235A, respectively). This result indicates that the highest affinity for SseK1 is achieved when the full WR-motif is present. Again, the results were compatible with the double mutant showing the lowest binding affinity. We then confirmed the differences in affinity compared to the WT peptide by performing competition STD NMR experiments. In binary samples containing SseK1 and equimolar concentrations of the TRADD$_{229-237}$ and one of the mutant TRADD peptides, none of the mutants was able to significantly displace the WT peptide, which demonstrates that modifications at the WR-motif impact negatively the affinity of the peptide for the enzyme (Supplementary Figs. 19, 20)

STD NMR data revealed that all the peptide ligands were recognized in solution. Hence, it is clear that differences in glycosylation of full-length FADD, TRADD, and GAPDH substrates by SseK1 and SseK2 are not due to differences in binding modes of their death domain sequences, but instead due to differences outside the binding site. In agreement with the similarity of binding modes of the short peptides detected by STD NMR, most of the sequence differences of SseK1 and SseK2 are likely in regions away from the binding site, including the HLH domain. Hence, differences in glycosylation specificity may be attributed to differences in the internal dynamics between the two enzymes at those distinct regions.

To test that hypothesis, we ran long (800 ns) Gaussian accelerated molecular dynamics (GaMD) simulations of SseK1 and SseK2[33]. Principal component analysis (PCA) showed that, in both cases, the motions of largest amplitude are indeed present around the HLH domain (Supplementary Fig. 7e, f), primarily

due to rotation of the HLH towards the binding site (Supplementary Fig. 7a, b). Noticeably, the simulations showed that SseK1 is significantly more flexible than SseK2 in the loop region connecting the HLH (Supplementary Fig. 7e, f). Additionally, significant differences were observed at the tip of the HLH, as SseK2 exhibited a substantial tilting motion towards the binding site (Supplementary Fig. 7c, d). These data reveal that there exist significant differences in the dynamics of the HLH domain between SseK1 and SseK2 that affect the HLH domain approach towards the substrate binding pocket, which could explain the differences in substrate specificity of these two enzymes and in turn, glycosylation.

To understand further the molecular basis of substrate peptide recognition we generated a ternary SseK2:UDP-GlcNAc:FADD$_{110-118}$ complex using an induced fit molecular docking protocol. Since this ternary complex was not accessible experimentally neither by X-ray crystallography nor by NMR spectroscopy, molecular modeling provides the only insight into the structure of the full complex in the presence of both the donor sugar nucleotide and the acceptor. Peptide structure prediction and NMR chemical shift indexing indicated that the peptide remained in its native helical conformation (Supplementary Fig. 8). Docking of FADD$_{110-118}$ was only possible using the SseK2 structure with the C-terminal lid in the open conformation, since the closed lid precludes access to the binding site. The resulting model was in good agreement with STD NMR data, with Trp112$^{FADD}$ and Arg113$^{FADD}$ in close proximity to the protein surface (Supplementary Fig. 9a, b). In addition, the sidechain of Arg113$^{FADD}$ was also found in close proximity to His260, Glu271, and Asn338, as observed in the docking of the acceptor Arg to the SseK2-UDP-GlcNAc crystal structure (Supplementary Fig. 9c and see below).

Furthermore, it was possible to graft the published 3D structure of the full-length FADD protein (PDB 3EZQ) into our SseK2:UDP-GlcNAc:FADD$_{110-118}$ model complex without any significant atomic overlap (Supplementary Fig. 10). We wanted to investigate whether the presence of the GlcNAc ring in the donor substrate might have an impact on the acceptor binding mode, and to analyze the dynamics of the full ternary SseK2:UDP-GlcNAc:FADD. We subjected this complex to a 500 ns Gaussian GaMD simulation. PCA showed that the most significant motions involved rotation and translation (Supplementary Fig. 11a, b) of both the SseK2 HLH and the FADD C-terminal α-helix towards one another. A distinct energy minimum was observed at short inter-helical distances (Supplementary Fig. 11c). At this minimum, a clear intermolecular complementarity was observed. In particular, electrostatic interactions were observed between Lys176$^{SseK2}$ and Asp175$^{FADD}$, and between Asp180$^{SseK2}$ and Arg166$^{FADD}$ (Supplementary Fig. 12a). Leu172$^{FADD}$ interacts with a hydrophobic patch defined by Val169$^{SseK2}$ and Leu170$^{SseK2}$. The α3-helix of FADD is highly negatively charged and interacts closely with Lys264, Arg348, and the manganese ion of SseK2 (Supplementary Fig. 12b). Our simulation suggests that Asp123 of FADD directly coordinates the manganese ion. Finally, along the 500 ns of the

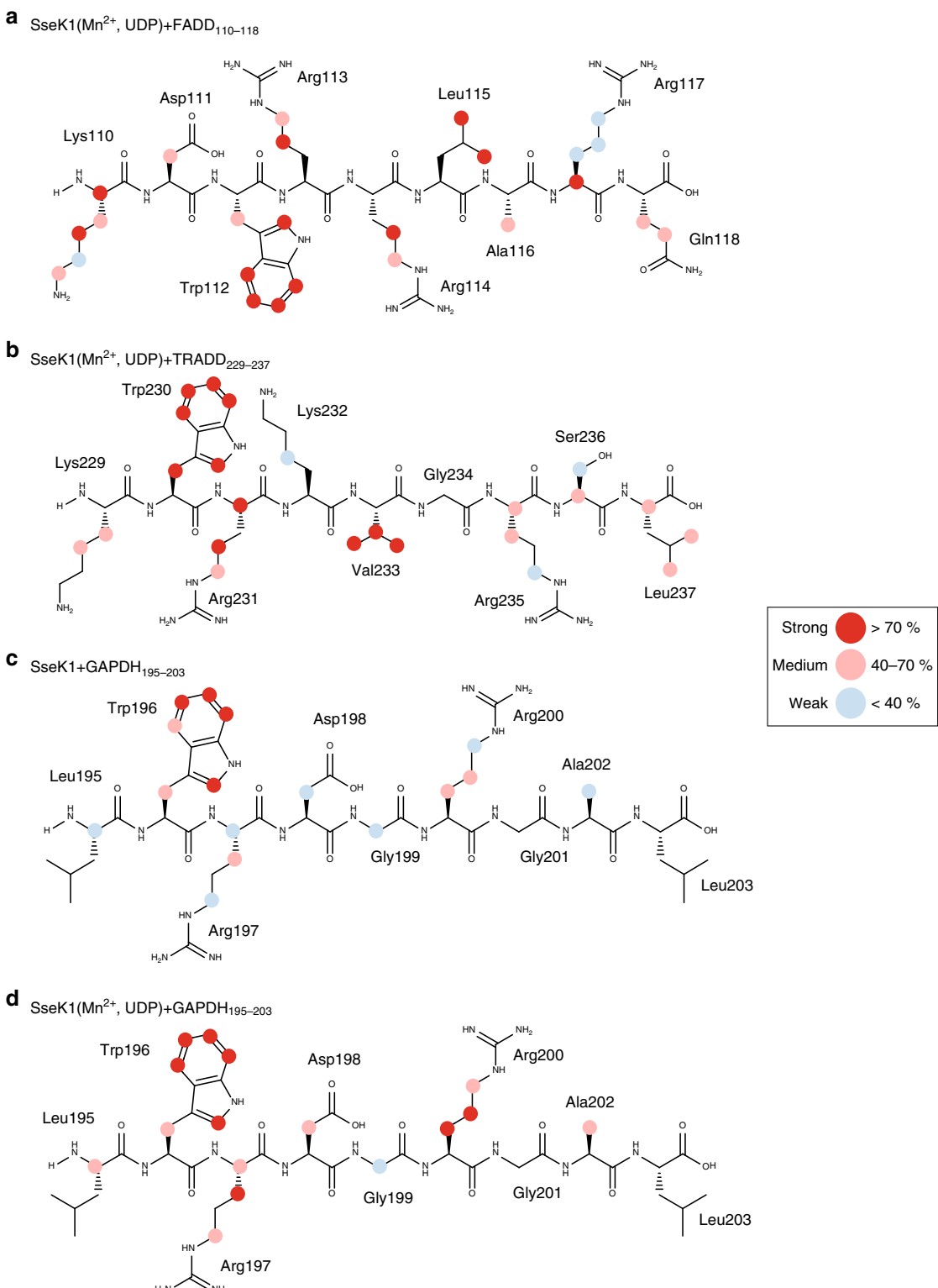

**Fig. 5** Binding modes of short peptide substrates. Binding epitope mappings of **a** FADD$_{110-118}$ **b** TRADD$_{229-237}$ and **c**, **d** GAPDH$_{195-203}$ peptides in the presence of 25 μM SseK1. Samples in **a**, **b** and **d** contained 25 μM Mn$^{2+}$, and 25 μM UDP. All STD intensities normalized against Hζ2 of the tryptophan. Colored circles represent magnitude of normalized intensities (blue:< 40%, pink: 40–70%, red:> 70%). Comparison of GAPDH$_{195-203}$ binding to SseK1, **c** in the absence, and, **d** in the presence of Mn$^{2+}$ and UDP, reveals a significant change in the binding mode of the substrate peptide upon addition of the cofactor and the nucleotide diphosphate. For STD NMR study of binding to SseK2 see Supplementary Fig. 5

GaMD simulations, significant conformational rearrangements of the arginine side chains in the $FADD_{110-118}$ region were observed (Supplementary Fig. 13a–d). On average, Arg117 of FADD is the residue from FADD showing the shortest distance to the $C_1$ anomeric carbon of UDP-GlcNAc, and is the only one simultaneously establishing close contacts with His260, Glu271, and the beta-phosphate of UDP-GlcNAc.

Together, these data provide evidence for specific recognition of FADD by SseK2 via interactions far from the active site, which may provide insight into differences in substrate glycosylation specificity of SseK1 and SseK2. These data also suggest that Arg117 is the best oriented residue from FADD for accepting the transferred glycosyl from the donor UDP-GlcNAc, in keeping with the role of this residue as the only acceptor site in FADD[5,6].

**Catalytic importance of HEN motif.** In the SseK2-UDP-GlcNAc crystal structure (chain D), Asn272 (corresponding to $Asn256^{SseK1}$) is closer to the $C_1$ anomeric carbon of UDP-GlcNAc (5.1 Å) as compared to Glu271 (5.6 Å). These data indicate that the Asn272 may play an important role in binding and catalysis. In addition, Glu271 is located at the entrance of the putative acceptor arginine-binding site. To obtain insight into the importance of these residues in substrate recognition, we conducted in-silico docking of the acceptor Arg to the SseK2-UDP crystal structure. It was observed that a closed conformation of SseK2 structure possesses a putative acceptor substrate binding site pocket that is connected to the anomeric carbon of GlcNAc. We used a closed conformation of SseK2 with a putative acceptor substrate binding site pocket connected to the anomeric carbon of GlcNAc for automated computational docking (Discovery Studio, Accelrys). As a result, the negatively charged pocket of the concave active site interacts with the positively charged guanidine group. In particular, Glu271, the β-phosphate of UDP, and His260 were located in a position suitable for hydrogen bonding (Supplementary Fig. 14a). All of these residues are highly conserved in the SseK and NleB families (Supplementary Fig. 2).

In the docking structure of SseK2, a single negatively charged residue ($Glu271^{SseK2}$) and an additional β-phosphate from UDP are in close contact to the guanidinium group of the acceptor Arg. The $Glu271^{SseK2}$ corresponds to the $Glu253^{NleB1}$ whose mutation to Ala did not inhibit NF-κB signaling[6], in agreement with the importance of this residue in glycosylation of the acceptor Arg. Further studies are required to determine its precise role either in catalysis or binding (see below).

In addition, $His260^{SseK2}$ is located near the guanidinium group of the acceptor Arg. A pH activity profile for both SseK1 and NleB1 revealed that NleB1 and SseK1 have an optimal pH between 6.0 ~ 8.0 and 6.0 ~ 8.5, respectively (Supplementary Fig. 14b). This highlights the potential role of histidine as a catalytic base because the $pK_a$ value of histidine is ~ 6.0. To test the role of these residues in substrate recognition and catalysis, we mutated His260, Glu271, and Asn272, which together form the 'HEN' motif. This motif is highly conserved at both the primary sequence and tertiary structure levels (Supplementary Fig. 15). Wild-type (WT) SseK1 and the HEN motif single mutants were overexpressed with TRADD in HEK293T cells. WT SseK1 inhibited cPARP production (Fig. 6a). As expected, the DxD-AxA double mutant, resulted in an increase in cPARP level. The HEN motif single mutants (His260, Glu271, and Asn272 in SseK2) led to an increase in the cPARP levels, a similar outcome to the DxD-AxA double mutant. When we observed the oligomerization form in a non-reducing gel, TRADD oligomer was detected in the mutant forms, suggesting that each mutant loses its glycosylation activity and fails to inhibit TRADD oligomerization.

NF-κB activity data in A549 cells correlated highly with the results from the PARP cleavage assay (Fig. 6b). Surprisingly, NF-κB levels increased more in H244A, E255A, and N256A than in the AxA mutant. Furthermore, we also investigated enzyme kinetics using the L-arginine substrate and the purified recombinant SseK1 and SseK2 (Fig. 6c, Supplementary Fig. 16a). The catalytic activities of the mutants (H244A/H260A, E255A/E271A, and N256A/N272A in SseK1/SseK2) decreased significantly compared to WT, though they were not essential for activity. The H244A/H260A and N256A/N272A mutants showed lower catalytic activity than the DxD-AxA double mutant.

We also studied the glycosylation activity of WT and HEN mutant enzymes in vitro, in cell culture, and in mouse infection experiments using *C. rodentium*. We first incubated NleB1, NleB2, SseK1, or SseK2 with FADD, TRADD, or GAPDH. We observed that, consistent with previous studies[7,18], WT NleB1, SseK1, and SseK2 glycosylated FADD (Fig. 6d). None of the point mutations in the HEN motif of any of the enzymes retained the ability to glycosylate FADD (Fig. 6d). Similar data were observed in studies of TRADD and GAPDH glycosylation (Fig. 6e, f). We also noticed that NleB1 and SseK1 are self-glycosylated, though the functional importance of this modification is unknown.

We also measured the ability of NleB1 and SseK1 to glycosylate TRADD and GAPDH when co-transfected into HEK 293T cells. We immunoprecipitated TRADD or GAPDH using an anti-FLAG antibody and then performed Western blotting for Arg-GlcNAc. Similar to our in vitro studies, we failed to observe any glycosylation of host substrates by any HEN mutation (Fig. 6g, h).

To extend these data, we conducted a series of mouse challenge experiments with *Citrobacter rodentium*. *C. rodentium* has only 1 copy of NleB, which functions similarly to EHEC NleB[18]. To evaluate whether the HEN motif of NleB is important to *C. rodentium* virulence, we deleted the *C. rodentium nleB* gene and then complemented this mutant with different EHEC *nleB1* and *nleB2* expression plasmids. We infected mice with *C. rodentium* Δ*nleB* strains expressing either WT NleB1 (Δ*nleB*/p*nleB1*) or the HEN mutants H242A, E253A, and N254A. Mice infected with Δ*nleB C. rodentium* showed an approximately 100-fold reduction in colonization magnitude after 14 days, as compared with WT *C. rodentium*, in support of previous findings[7] (Fig. 6i). While this mutant was fully complemented by expressing WT EHEC NleB1, none of the HEN mutants complemented the colonization defect (Fig. 6i). Additionally, neither WT EHEC NleB2 nor an NleB2 mutant in which all HEN amino acids were mutated to alanines complemented the colonization defect. These data demonstrate that the HEN motif is highly important for enzymatic activity and virulence.

**Proposed mechanism.** Three reaction mechanisms have been proposed for retaining GTs, namely $S_N2$, $S_Ni$, and orthogonal mechanisms. While the $S_N2$ mechanism involves a double-displacement reaction requiring a nucleophilic residue to form a covalent glycosyl intermediate[34], both the $S_Ni$ and the orthogonal mechanisms involve a single displacement reaction in which the β-phosphate of the nucleotide acts as the catalytic base[35–37]. However, the $S_Ni$ and the orthogonal mechanisms differ in their reaction profiles and the timing of bond formation and bond breakage[37]. These differences lead to a dissociative and associative transition state for the $S_Ni$ and the orthogonal mechanism, respectively[37]. Unlike the $S_Ni$ mechanism that has been probed extensively, the $S_N2$ and the orthogonal mechanism have never been demonstrated experimentally[34–37]. In our crystal structure, $Asn256^{SseK1}$ was located in a possible position for the back-side attack of the $C_1$ anomeric carbon of UDP-GlcNAc through an $S_N2$ reaction. However, our kinetic assay results showed that this

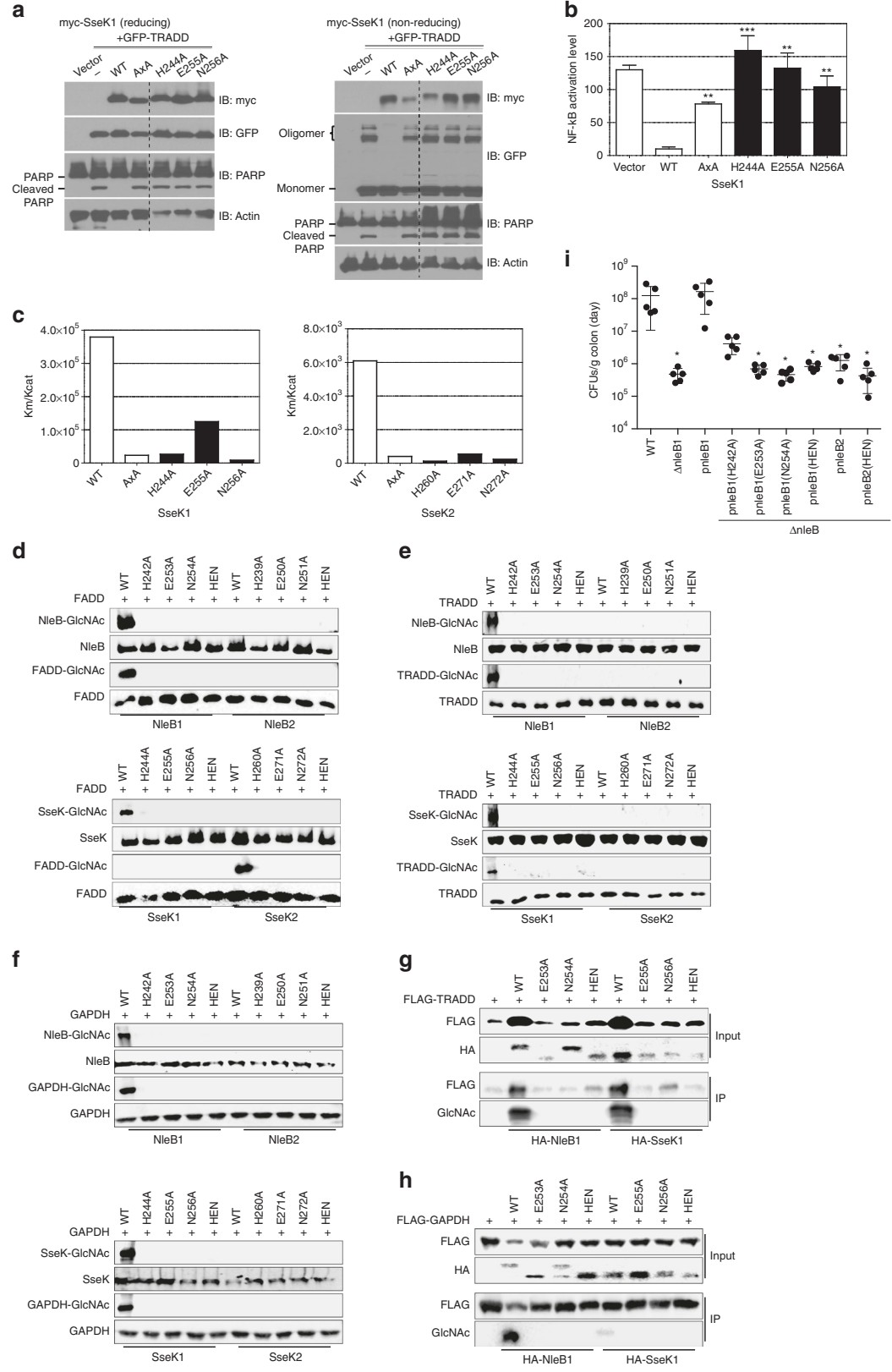

residue is not essential for catalysis (its mutation to alanine in both SseK1 and SseK2 reduced its $K_{cat}$ by 80 % as compared to the WT enzymes). This result is unexpected because other GTs completely lose their catalytic activity (650-fold to 23,000-fold decrease) when a potential nucleophile residue was mutated to a non-nucleophile residue[38,39]. It was proposed that Glu258 of

SseK3 acts as the nucleophile residue for an $S_N2$ reaction (corresponding to Glu255$^{SseK1}$ and Glu271$^{SseK2}$). However, the E255A$^{SseK1}$ and E271A$^{SseK2}$ mutants still possess an activity of 70 and 24%, respectively, ruling out its potential role as a nucleophilic residue. Additionally, our GaMD simulations of the grafted complex of SseK2 with FADD supported the relevance of Glu271

**Fig. 6** HEN motif plays a key role in NleB/SseK enzyme activity. **a** SseK1 mutants were generated and the cellular function in HEK293T cells was investigated. A non-reducing gel (right panel) was used to confirm the presence of the TRADD oligomer. Mutants in red represent mutations of residues proposed to be catalytically important. Data represent at least three repetitions. **b** The NF-κB level in A549-NF-κB luc cells was measured to check the enzymatic functions. Data represent the mean and standard deviation in triplicate. Multiple comparisons perform by one-way ANOVA followed by Turkey's Multiple Comparison Test (**$P < 0.01$, ***$P < 0.001$ compare to WT). **c** Enzyme kinetic assays of SseK1 and SseK2, respectively. **d** In vitro glycosylation of FADD by NleB1, NleB2 (top panel) and SseK1 and SseK2 HEN mutants (bottom panel). **e** In vitro glycosylation of TRADD by NleB1, NleB2 (top panel) and SseK1 and SseK2 HEN mutants (bottom panel). **f** In vitro glycosylation of GAPDH by NleB1, NleB2 (top panel) and SseK1 and SseK2 HEN mutants (bottom panel). **g** Glycosylation of TRADD after co-transfection with either NleB1 or SseK1 (WT and HEN mutants) in HEK293T cells. FLAG-TRADD was immunoprecipitated and then immunoblotted using an anti-Arg-GlcNAc antibody. **h** Glycosylation of GAPDH after co-transfection with either NleB1 or SseK1 (WT and HEN mutants) in HEK293T cells. FLAG-TRADD was immunoprecipitated and then immunoblotted using an anti-Arg-GlcNAc antibody. **i** Colonization (log10 CFUs/g colon) of indicated *C. rodentium* strains (14 days post-gavage) in C57BL/6 J mice ($n = 6$). Asterisks indicate significantly different colonization magnitude as compared to WT; Kruskal-Wallis test. Uncropped blots are shown in Supplementary Figs. 21 and 22

and His260 in Arg recognition, as they established stable interactions with the guanidinium group of Arg117 of FADD (Supplementary Fig. 13b, c). In turn, the center of mass of the guanidinium of Arg117$^{FADD}$ was 4 Å from the anomeric carbon of the GlcNAc residue of the donor substrate in the 500 ns simulations, in a location compatible with a front face mechanism for glycosyl transfer with retention of anomeric configuration (see below; Supplementary Fig. 13e). Hence, our results are more consistent with an $S_Ni$ mechanism in which the β-phosphate of the UDP formed in the reaction acts as the catalytic base to fully activate the acceptor Arg, but are inconsistent with the $S_N2$ based double-displacement mechanism proposed earlier (Supplementary Fig. 17). Note that we cannot rule out the possibility of these enzymes following an orthogonal mechanism, although as mentioned previously, this mechanism has never been probed experimentally.

## Discussion

The overall crystal structures of SseK1 and SseK2 from *S. typhimurium* as well as NleB2 from enteropathogenic *E. coli* are similar, revealing high identities between the amino acids at the active binding site level. These enzymes glycosylate the guanidinium moiety of Arg residues, which are residues with poor nucleophilic character because electrons on this moiety are partially delocalized. In addition, we identified several characteristics in the active site that are compatible with an $S_Ni$ mechanism and with the role of HEN motif residues in catalysis: (a) we experimentally demonstrate by NMR that these enzymes are retaining GTs; (b) donor-substrate-mediated fixation of the C-terminal lid could shield the active site from the hydrophilic environment, avoiding unwanted hydrolysis of UDP-GlcNAc, -a behavior reported in other retaining GT-A fold GTs such as *Legionella pneumophila* glucosyltransferase[27] and the toxin B;[40] (c) mutations of the HEN motif lead to a significant reduction in glycosylation both at in vitro and in vivo levels, implying that these residues affect enzyme catalysis. Based upon our kinetic, structural, and computational studies, we suggest that His and Glu might improve the poor nucleophile character of the acceptor Arg guanidinium moiety to facilitate catalysis; and (d) based on our MD simulations, the acceptor Arg is facing the anomeric carbon, which is compatible with a front face mechanism for glycosyl transfer with retention of anomeric configuration.

Each HEN residue of SseK3 is on the β-strand, but in the case of SseK1, SseK2, and NleB2, the HEN residues are located on the loop structure. This has implications for the differences in the activity of the enzymes due to the HEN motif because the loop structure is more flexible than the β-strand. In addition, there are differences in the regulation of the C-terminal lid domain between SseK1, SseK2, and SseK3. The arginine on the C-terminus participates in an interaction with UDP-GlcNAc in

SseK2 and SseK3, but not in SseK1. Based on this result, the departure of the leaving group is likely to be easier, due to a weaker enzyme-substrate interaction network in SseK1, than in SseK2 and SesK3. We suggest that this may also be a reason for the differences in enzyme activity in the SseK family.

Although the tertiary structure of NleB1 is likely similar to NleB2, SseK1, and SseK2, a previous study has reported that NleB2 has a lower activity than NleB1 on the same target (TRADD)[6]. In this study, we also demonstrate that the enzymatic activity of SseK1 is ~ 62 times higher than that of SseK2 based on enzyme kinetic assays. The substrate specificities of the NleB/SseK family of *C. rodentium*, *Escherichia coli* and *Salmonella enterica* are different[18]. Based on these differences, we can infer from the crystal structures that these discrepancies between orthologs of this family might be attributed partially to the HLH domain, which is a structural feature not present in other GTs (Supplementary Fig. 18a). The amino acids of this domain are not conserved and are structurally flexible (Supplementary Fig. 18b), as confirmed by the long GaMD simulations (Supplementary Fig. 7). The flexible HLH domain is close to the active site, which indicates this domain may be involved in the recognition of the acceptor protein substrates containing death domains. In fact, GaMD simulations of the grafted SseK2:UDP-GlcNAc:FADD complex show that the HLH domain interacts directly with FADD C-terminal α-helix through side chain complementarity (Supplementary Fig. 12). These data support the role of the HLH domain in the recognition of the acceptor protein substrates containing death domains.

STD NMR spectroscopy revealed that, in contrast to their enzymatic activity profile, both SseK1 and SseK2 interact with all short peptides from FADD, TRADD, and GAPDH (Fig. 5, Supplementary Fig. 5). Each of these peptides contains a conserved WR-motif, which forms a key structural requirement for binding to the enzymes, as revealed by STD NMR spectroscopy and molecular modeling. Therefore it appears that recognition of death domains may be due to complementarity with the active site, whilst distinction between different death domain-containing proteins is mediated through interactions far from the active site. Overall our data provide compelling evidence of the molecular basis for Arg glycosylation, the differences in substrate specificity among orthologs, and will provide a framework for the design of pan-NleB/SseK inhibitors targeting enteric pathogens.

## Methods

**Protein purification.** SseK1(21-336) gene was generated and amplified by PCR from synthesized DNA and cloned into a modified pET28a (Novagen) in which the thrombin cleavage site was replaced with a tobacco tech virus (TEV) protease cleavage site. SseK2 (34-348) and NleB2 (1-316) gene were generated and amplified by PCR from *Salmonella typhimurium* (strain SL1344), *Escherichia coli* O145:H28 (strain RM12581) respectively and cloned into the pVFT3S vector (Korean patent 10-0690230), which has 6xHis-thioredoxin (Trx) and TEV protease cleavage site (Supplementary Table 2, Supplementary Table 3). PCR-based site-directed

mutagenesis was employed to generate various point mutations. Complete amino acid sequences are shown in Supplementary Table 4.

Each sub-cloned plasmid was transformed into *E. coli* BL21(DE3) (Novagen) and grown in high salt Luria-Broth medium. When the O.D$_{600}$ reached 0.6 ~ 0.8, the temperature was decreased to 17 °C and the culture was induced with 0.3 mM IPTG (isopropyl 1-thio-ß-D-galactopyranoside). After 16 h incubation, each protein was purified using nickel-affinity chromatography. Cell was lysed using lysis buffer (20 mM Tris-NaCl pH 7.5, 300 mM NaCl, 30 mM imidazole, 10% glycerol) and the proteins were eluted using an elution buffer of 300 mM imidazole in lysis buffer. Thereafter, the TEV recognition site was cleaved using TEV protease. After desalting to 20 mM Tris-HCl (pH7.5), 50 mM NaCl, each protein was loaded into an anion-exchange chromatography column (Hitrap-Q, GE healthcare) and then gel-filtration chromatography (Superdex-200, GE healthcare) was carried out in SEC buffer (25 mM HEPES-NaOH [pH7.5], 300 mM NaCl).

Selenomethionine substituted NleB2 was prepared using *E. coli* B834 (Novagen) and cultured in M9 minimal media supplemented with glucose, amino acids, and L-selenomethionine (Calbiochem). Expression condition and purification method were the same as for native NleB2.

NleB/SseK genes were also cloned into pET42a and then subjected to site-directed mutagenesis. GAPDH and TRADD were cloned into pET28a. FADD was cloned into pET15b. Proteins were purified after their overexpression in *E. coli* BL21(DE3) using Ni-NTA agarose. NleB/SseK genes were sub-cloned into pCMV tag 2a, pCMV Myc or (HA tag vector) for mammalian expression. TRADD gene was cloned into pCMV tag 2a or pEGFP N1.

**Crystallization and data collection**. Purified SseK1 and SseK2 were concentrated to 25 mg/mL and co-crystallized with 5 mM UDP and 5 mM MnCl$_2$. Initial crystallization screening was conducted by using Mosquito robot (TTP Labtech.) and single, appropriate size of crystals appeared in 0.1 M Bis-Tris propane-HCl (pH 7.0), 1.0 M ammonium citrate tribasic [pH 7.0] and 0.1 M Bis-Tris [pH 6.5], 26% (w/v) PEG3350, respectively. SseK2 apo crystals appeared 0.1 M HEPES-NaOH [pH 7.5], 0.1 M sodium acetate, 24% (w/v) PEG4000 and for revealing UDP-GlcNAc bound SseK2 structure, 5 mM MnCl$_2$, 5 mM UDP-GlcNAc were added into SseK2 apo crystal drop. 13 mg/mL of purified NleB2 formed crystals at 0.8 M LiCl$_2$, 0.1 M Tris-HCl [pH8.5], 8% (w/v) PEG3350. The Se-Met derivative of NleB2 crystallized under the same conditions. 20 % ethylene glycol was added as a cryoprotectant to each crystal solution and flash frozen in liquid nitrogen.

All of the crystal diffraction experiments were carried out at Photon factory (KEK, Tsukuba, Japan). UDP or UDP-GlcNAc bound SseK2 and UDP bound SseK1 were diffracted at BL-5A, BL-1A beamline, respectively. Native NleB2 and Se-Met derivative NleB2 crystal was diffracted at BL-17A.

**Structure determination and refinement**. Diffraction data sets were processed and scaled with the programs imosflm[41] and Aimless from the CCP4 program suite. The phasing information was solved by SAD method from Se-Met derivative NleB2 crystal using AutoSol program and the other proteins were solved by molecular replacement using NleB2 structure. MOLREP, REFMAC5, and COOT were used for molecular replacement, structure refinement, and further modeling, respectively. All figures were prepared using PYMOL.

**Peptide assignment and STD NMR**. All experiments were performed at 288 K on a Bruker Avance III 800 MHz spectrometer equipped with a 5-mm TXI 800 MHz H-C/N-D-05 Z BTO probe. FADD$_{110-118}$ and GAPDH$_{195-203}$ (Genscript) samples were prepared at 1 mM in 90% H$_2$O/ 10% D$_2$O and assigned using standard COSY (cosydfesgpph), TOCSY (mlevphpr), and $^1$H-$^{13}$C HSQC (hsqctgpsp) experiments. Apo-enzyme samples were prepared with 1 mM peptide and 25 µM enzyme in either 25 mM Tris-d$_{11}$ (SseK1) or 10 mM PBS (SseK2); both at pH 7.4 in D$_2$O. Holoenzyme samples were prepared in the same way, with the addition of 25 µM MnSO$_4$ and 25 µM UDP. The residual water signal was used as a reference for chemical shifts. STD NMR experiments were performed using a train of 50 ms Gaussian pulses (0.4 mW, B$_1$ field strength 78 Hz) applied on the f2 channel at either 0 ppm (on-resonance) or 40 ppm (off resonance). A spoil sequence (2 trim pulses of 2.5 and 5 ms followed by a 40 % z-gradient applied for 3 ms at the beginning of the experiment) was used to destroy unwanted x,y-magnetization from previous scan and a spinlock (1.55 W, 40 ms) was used to suppress protein signals (stddiff.3). The saturation time (d20) was set to 2 s and the recycle delay (d1) was set to 5 s.

**Configuration of GlcNAc in the glycosylated peptide**. Samples for peptide glycosylation were prepared by adding either 7.5 mM FADD$_{110-118}$ or 7.7 mM GAPDH$_{187-203}$ to 50 mM UDP-GlcNAc, 40 µM SseK1, and 2 mM MnCl$_2$ in 25 mM Tris pH 7.5, allowing the reaction to proceed for 24 h at 37 °C. The resulting glycopeptide was purified from the enzyme by using an Amicon® Ultra 10 K device. NMR experiments for the GAPDH$_{187-203}$ sample were then performed at 298 K, and consisted of a decoupled $^1$H-$^{13}$C HSQC (hsqcetgpsi), and TOCSY with water suppression (mlevgpph19) at 800 MHz, and a Perfect-CLIP-HSQC[42] at 500 MHz (with a digital resolution of 1.6 Hz, to determine the $^1J_{C,H}$ coupling of the anomeric carbon of the transferred GlcNAc residue). The HSQC recycle delay was 1.5 s. For the TOCSY, the recycle delay was 2 s and the mixing time was 80 ms. For the

FADD$_{110-118}$ sample, a decoupled $^1$H-$^{13}$C HSQC was recorded as above, in a Bruker Avance I 500 MHz spectrometer, equipped with a triple resonance indirect detect TXI probe with Z-gradients.

**Molecular docking calculations for guanidine-SseK2**. UDP-bound SseK2 structure and guanidine was loaded to Discovery Studio 4.0 and the possible binding site was set (Radius = 10 Å, XYZ = 21.197,−9.134, 12.013). The algorithm was taken from the CHARMm protocol and the best score was selected (-CDOCKER ENERGY = 18.192, -CDOCKER INTERACTION ENERGY = 18.097)

**Molecular docking calculations for FADD-SseK2**. Crystal structures of SseK1, SseK2, and FADD (PDB 3EZQ) were imported into Schrödinger Maestro[43] and prepared with the Protein Preparation Wizard[44]. All buffer atoms and non-bridging waters were removed. Protons were then added to the model, using PROPKA to predict the protonation state of polar sidechains at pH 7[45]. The hydrogen-bonding network was automatically optimized by sampling asparagine, glutamine, and histidine rotamers. The model was then minimized using the OPLS3[46] force field and a heavy atom convergence threshold of 0.3 Å.

A model of the FADD$_{110-118}$ peptide was created by truncation of the FADD crystal structure. Conformers were generated in MacroModel by torsional sampling with the OPLS3 forcefield, constraining all backbone atoms. Redundant conformers were eliminated using an RMSD cutoff of 0.5 Å. Any conformer with an energy 5 kcal mol$^{-1}$ greater than the lowest energy structure was also eliminated. Resulting conformers were then minimized using the conjugate gradient method, converging on a threshold of 0.05 kcal mol$^{-1}$. Docking of FADD$_{110-118}$ to SseK2 was then performed using Glide[47,48]. A cubic grid, suitable for peptide docking, was generated. It was centered on UDP-GlcNAc, with an outer box length of 45 Å and an inner box length of 40 Å. To account for flexibility, van der Waals potentials of all receptor and ligand atoms were scaled by 0.5. All ligand conformers were docked to the receptor using rigid sampling with the SP algorithm. The resulting complexes were then clustered by heavy atom RMSD to eliminate redundant poses, keeping the structure closest to the cluster centroid from each cluster. All sidechains within 5 Å of the ligand were then optimized before minimizing using Prime[49]. A second round of docking was performed, as described above, on the new receptor structures. The resulting complexes were clustered by heavy atom RMSD, and the lowest energy representative structure was chosen for analysis.

A model of SseK2 in complex with full length FADD was generated by aligning the backbone atoms of residues 110-118 in the full-length structure to the backbone of the docked FADD$_{110-118}$ structure. Prime optimization and minimization within 5 Å of the contact surface was used to eliminate an atomic overlap.

**Molecular dynamics**. UDP charges for use with UDP-GlcNAc were derived using the RESP fitting method implemented on the RED server[50]. The UDP fragment was generated by replacing the GlcNAc with a methyl group. In accordance with GLYCAM[51], the HF/6-31G* level of theory was used with a weight factor of 0.01 and all aliphatic protons were constrained to a charge of 0. The total charge of the UDP fragment was set to −2. The charge of the methyl group was set to 0.194 before removing to give a final fragment with net charge −2.194, in keeping with the modularity of GLYCAM.

Molecular dynamics simulations of SseK1, SseK2, and the SseK2:FADD complex were performed using the Amber PMEMD software[52]. Protein atoms were parameterized using the Amber ff11SB forcefield and the Mn$^{2+}$ ion was modeled using 12-6-4 LJ-type parameters (Amber ions234lm_1264_tip3p). UDP-GlcNAc was parameterized with GLYCAM 06j and GAFF. Each system was solvated in a truncated octahedral box of TIP3P water, with at least 10 Å between the solute and the edge of the box, before neutralizing with Na$^+$ ions. The system was minimized using the conjugate gradient algorithm, converging on a threshold of 10$^{-4}$ kcal mol$^{-1}$ A$^{-1}$, first with 20 kcal mol$^{-1}$ A$^{-2}$ restraints on solute atoms, before repeating with no restraints. The system was slowly heated to 310 K over 500 ps (NVT), before equilibrating the pressure to 1 atm (NPT) over a further 500 ps. In both cases with 20 kcal mol$^{-1}$ A$^{-2}$ restraints were used on solute atoms. These restraints were then slowly released over 800 ps before performing Gaussian accelerated molecular dynamics (GaMD) simulations (800 ns SseK1/SseK2, 500 ns SseK2:FADD complex), as implemented in AMBER, using a boost potential on both the dihedral and total potential energies. Here, the simulation was split into 4 distinct stages. First, conventional dynamics were run for 2 ns to automatically calculate an initial boost potential. The calculated boost potential was then applied and fixed for 400 ps before allowing it to adapt for a further 5.6 ns. The resulting boost potential was then fixed before performing production dynamics for 800 ns (SseK1/SseK2) or 500 ns (SseK2:FADD complex), saving coordinates every 100 ps. In all cases, the SHAKE algorithm was used to restrain all bonds involving hydrogen, allowing for a time step of 2 fs. A Langevin thermostat was used with a collision frequency of 5 ps$^{-1}$ and the barostat used an isotropic Berendsen algorithm with a relaxation time of 1 ps. In all cases, periodic boundary conditions were used, using the particle mesh Ewald to calculate electrostatics.

**Cell culture**. Human embryonic kidney (HEK) 293T (ATCC, ATCC® CRL-3216™) and A549 NF-κB luciferase cells (Panomics, RC0002) were cultured in DMEM

supplemented with 10% fetal bovine serum (FBS, Cellgro), 100 U/ml penicillin, 100 µg/ml streptomycin, and 2 mM L-glutamine at 37 °C in 5% $CO_2$.

**Western blot analysis and immunoprecipitation.** HEK293T cells were transfected with various combinations of plasmids using Lipofectamine 2000 (Invitrogen) as specified by the manufacturer. Cells were washed with phosphate-buffered saline (PBS) and lysed in 1 × RIPA buffer (GenDEPOT, Barker, TX, USA) containing 150 mM NaCl, 1% Triton X-100, 1% deoxycholic acid sodium salt, 0.1 % sodium dodecyl sulfate (SDS), 50 mM Tris-HCl (pH 7.5), 2 mM EDTA, and a protease inhibitor cocktail. Whole cell lysates (WCLs) were centrifuged at 13,000 rpm for 10 min at 4 °C. To detect TRADD oligomerization, WCL were separated using a non-reducing sample buffer. About 30 µg of proteins were separated by SDS-PAGE and transferred to nitrocellulose membranes (GE Healthcare, Little Chalfont, UK). Non-specific binding was blocked, and anti-GFP (1:3000, Santa Cruz, CA, USA, SC-8334, SC-9996), anti-actin (1:5000, Cell Signaling Technology, Danvers, MA, USA, #4967), and anti-PARP (1:1000, Cell Signaling Technology, Danvers, MA, USA, #9542) anti-c-Myc (1:3000, Invitrogen, Camarillo, CA, USA, 13-2500) antibodies were used as primary antibodies. After washing, membranes were probed with the HRP-conjugated secondary antibody for 1 h. Enhanced chemiluminescent substrate (GenDEPOT, Barker, TX, USA) was used for visualization.

Immunoprecipitation was performed with 15 µl of dynabeads protein G (Invitrogen, Camarillo, CA, USA). The beads were washed and incubated with 1 µg of the antibody for 1 h at RT. The beads were incubated with 300 µg WCL overnight at 4 °C after washing. Samples were separated using SDS-PAGE for immunoblotting. UDP-GlcNAc (1 mM) and $MnCl_2$ (5 mM) were added to recombinant SseK1 and incubated at 37 °C for 1 h. The same amount of wild type and auto-glycosylated SseK2 were loaded into 15% SDS-PAGE gel and anti-GlcNAc antibody (1:5000, CTD110.6, Santa Cruz, CA, USA, #sc-59623 used to detect GlcNAcylated arginine).

**NF-κB luciferase assay.** A549 cells (Panomics, RC0002) stably expressing NF-κB were transfected with a mixture of pNL1.1.TK[*Nluc*/TK], GFP-TRADD and various Myc-SseK1 plasmids. pNL1.1.TK[*Nluc*/TK] was used for transfection control. After 24 h, the cells were treated with 20 ng/ml TNF-α for 6 h. Luciferase assay was performed using Dual-Luciferase Reporter Assay system (Promega, E1910). Briefly, cells were lysed with Luciferase Cell Culture Lysis Reagent and Luciferase Assay Reagent II was added to measure the luciferase activity. Stop & Glo Reagent was added into tube to quench firefly luciferase activity, and nanoLuc luciferase activity was measured using a luminometer.

**Glycosyltransferase kinetics assay.** Recombinant SseK1, SseK2 and point mutant proteins were prepared as described and GT kinetics were measured using UDP-Glo™ Glycosyltransferase Assay kit (Promega, #V6961) by manufacturer's instruction. Enzyme reaction buffer (ERB) was prepared as 25 mM Tris-HCl [pH 7.5], 50 mM NaCl, 4 mM $MnCl_2$, 1 mM DTT and reaction was eliminated by using the nucleotide detection buffer. For preparing the acceptor-substrate, L-arginine (Duchefa BIOCHEMIE, > 98.5% purity) was dissolved in ERB. Synthetic peptides of GAPDH (Genscript, > 90% purify) was purchased and dissolved in ERB. White 96-well plates (ThermoScientific) were used for luminescence assay and the plate was read by using luminometer (VictorX5, PerkinElmer). Kinetics parameters were calculated using GraphPad Prism5 ver.5.03 software. The points represent an average of two samples and error bars represent mean ± S.D. The final specific activity of the transfer reaction was corrected considering the hydrolysis reaction, which was performed using SseK1/SseK2, UDP-GlcNAc, and $MnCl_2$.

**Mouse infections.** All animal experiments were performed according to Institutional Animal Care and Use Committee-approved protocols (Animal Welfare Assurance #3647) and conducted as previously described[7]. Female BALB/c mice were obtained from the Jackson Laboratory, housed in microisolator cages, and provided with food and water ad libitum. *C. rodentium* ΔnleB was electroporated with pFLAG-CTC plasmids expressing EHEC *nleB1* or *nleB2* genes. Mice were challenged with $1*10^8$ CFUs of each strain and observed twice daily for 14 days. Colon samples were dissected after euthanasia, homogenized, serially diluted in PBS, and then plated on MacConkey agar to enumerate bacteria.

**pH-dependent GT activity test.** Glycosylation reactions were performed as described previously[18]. Briefly 200 nM of enzyme was mixed with 1 µM substrate in the presence of 50 mM Tris-HCl pH 7.4, 1 mM UDP-GlcNAc, 10 mM MnCl2, and 1 mM DTT for 2 h at room temperature. Samples were subjected to Western blot analysis using an anti-Arg-GlcNAc antibody (Abcam). The pH-dependence of glycosylation was assessed under similar conditions except that the Tris-HCl was replaced with McIlvaine buffers.

**Isothermal titration calorimetry.** ITC experiments were conducted in a MicroCal VP-ITC (MicroCal) device. After size exclusion chromatography (SEC) in Tris-HCl [pH 7.5], 150 mM NaCl, and 1 mM $MnCl_2$, sample fractions corresponding to the single UV$_{280}$ peak were collected and concentrated to 0.2 mM by using 10 kDa

cut-off Amicon tubes. Ligands (4 mM) were dissolved in the same SEC buffer and were titrated to variant SseK proteins at 25 °C. Binding stoichiometry, enthalpy variation, entropy variation, dissociation constant and Chi-square values were calculated using MicroCal Origin software.

## Data availability

Coordinates and structure factors have been deposited in the Protein Data Bank under the accession codes 5H5Y (NleB2 X-ray structure), 5H60 (UDP bound SseK1 X-ray structure), 5H61 (SseK2 X-ray structure), 5H62 (UDP-bound SseK2 X-ray structure) and 5H63 (UDP-GlcNAc bound SseK2 X-ray structure). Other data are available from the corresponding authors upon reasonable request.

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

## Acknowledgements

We thank the staff scientists for assistance at the beamline 5A and 1A of the Photon Factory and the beamline 5C and 7A of Pohang Light Source. This work was supported by Grants from the National Research Foundation of Korea (NRF) funded by the Korean government (MEST) (NRF- 2016R1A2B2013305, 2016R1A5A1010764, 2014R1A4A1008625, 2017M3A9F6029755 and 2017R1A2B3006704), the Strategic Initiative for Microbiomes in Agriculture and Food funded by Ministry of Agriculture, Food and Rural Affairs (918012-4) and Brain Korea 21 PLUS Project for Medical Science. This work was also supported by Grants AI093913 and AI127973 from the National Institutes of Health (to P.R.H.). We also thank ARAID and MEC (CTQ2013-44367-C2-2-P, BFU2016-75633-P to R.H-G.), and the DGA (group number E34_R17) for financial support. S.M. acknowledges a postgraduate studentship from the School of Pharmacy of the University of East Anglia. J.A. and S W. acknowledge funding from BBSRC through a research grant (BB/P010660/1) and a DTP PhD studentship, respectively. We thank Dr Ridvan Nepravishta for helpful discussions and technical assistance with the NMR studies. We thank prof. Jihyun F. Kim (Yonsei University) for providing us with DNA of NleB1 and 2.

## Author contributions

J.B.P. designed protein constructs, collected the diffraction data, and solved the structures. Y.H.K. performed NF-κB and western blot assays. J.B.P., Y.H.K., S.-H.J., J.A., J.-S.S., R.G.-H., P.R.H. and H.-S.C. wrote the manuscript. Y.K.Y. and J.Y.K. expressed, purified, and crystallized each protein. J.-S.S., J.W.C. and H.-S.C. designed, directed, and supervised the crystallography experiments. A.A.G.-G. prepared samples for NMR experiments. S.E.Q., M.W., and M.P.H. conducted *C. rodentium* virulence assays. S.E.Q. and M.W. performed glycosylation assays. S.W. and S.M. carried out the NMR experiments. S.W. carried out the molecular dynamics simulations. S.W., S.M. and J.A. analyzed the NMR and molecular dynamics data.
