## [Peer Review File · Nature Communications]

Reviewers' comments:

Reviewer #1 (Remarks to the Author):

Park and co-workers report crystal structures of the bacterial effectors SseK1 and SseK2 from *Salmonella typhimurium* SL1344, and that of NleB2 from the *Escherichia coli* O145:H28, all involved in the glycosylation of arginine residues in host proteins. Specifically, the authors present (1) one crystal structure of SseK1 in complex with UDP at 3.64Å; (2) three crystal structures of SseK2: the unliganded form at 1.86Å, a binary complex with UDP at 1.66Å, and a binary complex with the sugar donor UDP-GlcNAc at 1.92Å; and (3) one crystal structure of the unliganded form of NleB2 at 2.3Å. SseK1, SseK2 and NleB2 display a typical GT-A fold. By using a combination of X-ray crystallography, STD-NMR, enzyme kinetics, molecular dynamics simulations, and in vivo experiments, the authors propose a plausible model for substrate/s binding and specificity, and catalytic mechanism. It is worth noting that the crystal structure of SseK3 was previously/very recently determined Esposito et al. *J. Biol. Chem.* 2018 Apr 6; 293(14):5064-5078. However, the current work reported by Park and colleagues, is sufficiently novel and merits its publication in *Nature Communications*.

Some questions to be addressed and suggestions for improvement are listed below:

1. page 4, line 69: "GTs are classified according to their folding as GT-A, GT-B, or GT-C.". A GT-D fold has been recently proposed. Please see:

Zhang H, Zhu F, Yang T, Ding L, Zhou M, Li J, Haslam SM, Dell A, Erlandsen H, Wu H. The highly conserved domain of unknown function 1792 has a distinct glycosyltransferase fold. *Nat Commun.* 2014 Jul 15; 5:4339.

Please could you comment on it?

2. page 4, line 70: Most GT-A enzymes possess an DXD signature in which the carboxylates coordinate a divalent cation and/or a ribose. However, examples do now exist of enzymes from this fold family that do not possess the DXD signature. Please refer to:

Lairson LL, Henrissat B, Davies GJ, Withers SG. Glycosyltransferases: structures, functions, and mechanisms. *Annu Rev Biochem.* 2008; 77:521-55.

Therefore, please modify the sentence accordingly.

3. page 4, line 71: The donor substrates include sugar-linked nucleotide diphosphates that also interact with the cofactor

4. page 4, line 79: please define non-LEE.

5. page 5, line 94: 'The T3SS effectors SseK1 and SseK2 from *Salmonella typhimurium* SL1344 are NleB orthologs that behave as NleB1-like GTs, although they differ in protein substrate specificity'. I would suggest the authors to introduce this information in the abstract. In addition, 'specificity' by 'specificity'.

6. page 6, line 124: Please replace by 'We therefore provide compelling experimental evidence by NMR that the transfer of GlcNAc by SseK1 follows a retaining mechanism.' In addition, explain why you consider that the SseK2 is a retaining GT.

7. page 66, line 128: Please replace the title 'GT overall architecture' by 'Overall architecture of SseK1, SsK2 and NleB2 GTs'

8. page 6, line 130 and 136: Instead, please state the form in which the crystal structure was obtained for each protein. For example: NleB2 was solved in its unliganded form. SseK1 was solved in complex with UDP. SseK2 was solved in its unliganded form and in complex with UDP and UDP-GlcNAc.
9. page 6, line 132: 'the N-terminus of SseK1 and SseK2 were truncated, respectively.' Please explain the reason of that.
10. page 7, line 143: 'because their datasets were obtained at a higher resolution.'" And the authors have three different snapshots of the active site.
11. page 7, line 145: 'root-mean-square deviation value of the structure is ~2.0 Å based on DALI pairwise comparison'. Please include the r.m.s.d. values, Z-score as well as number of compared residues.
12. page 7, line 146: 'Most GT-A fold GTs are single domain proteins that contain a Rossmann-like fold.', 'which has two abutting $\beta/\alpha/\beta$ Rossmann-like domains ($\beta3-\alpha2-\beta4-\alpha3-\beta5$)', 'SseK2 can be divided into three types of sub-domains, namely'. Please clarify. I would suggest to include residue numbers to support the 'domain' and/or subdomains description.
13. page 7, line 156: 'However, in the complex structure with the UDP or UDP-GlcNAc, the substrate leads to an unambiguous electron density map for the C-terminal lid domain,.'. Please specify the protein complexes.
14. page 8, line 186: "In the SseK3 structure, SseK2-like π - π stacking interaction is conserved and Phe190SseK3 and Trp52SseK3 (corresponding to Phe203SseK2 and Trp65SseK2) participate in an interaction with the uridine group in the same orientation." A good opportunity to introduce a Figure 3 panel showing the arrangement of these aromatic residues into the SseK3 active site.
15. page 9, line 214: What is the experimental evidence that support the substrate binding/product release order for SseK1, SseK2, SseK3 or NleB2? I agree with the authors that in most of the cases, GTs follow an order mechanism with the donor substrate binding first. However, in the absence of experimental enzymatic/kinetic data, the authors should be cautious with the interpretation of the structural data. A statement should be added accordingly.
16. page 9, line 229: 'Based on these data, we suggest that the conformation of the C-terminal domain is highly similar between SseK2 and SseK3.' Please remove this sentence.
17. page 10, line 245: 'STD NMR experiments of SseK1 and SseK2 in the presence of the peptides FADD110-118, TRADD229-237, and GAPDH195-203 showed that all three peptides bound to both enzymes in the presence and absence of Mn^{2+} and UDP'. Please rephrase the sentence.
18. page 10, line 254: 'notion that that a' by 'notion that a'
19. page 13, line 323: I would suggest the authors to introduce here a section focused into the 'Catalytic mechanism of SseK1, SseK2 and NleB2'. In that context, please move/introduce the entire 'Anomeric configuration of glycosylated peptides' section here, and renumber Figure 1 accordingly. After the discussing the catalytic mechanism, the authors could introduce the section 'Catalytic importance of HEN motif'.
20. page 15, line 394. 'Two reaction mechanisms, SN_2 and SN_i , have been proposed for retaining GTs.' Please see:

Evans SV, Fyles TM.

Please, could you comment on this?

21. page 17, line 432: 'these residues are important in catalysis' by 'these residues impact enzyme catalysis'.

22. page 18, line 470: I would remove the last paragraph 'In summary...'.

23. page 19, line 480: The complete amino acid sequence of each protein construct should be included as a Supplementary Figure.

24. Supplementary Table 1.

Data Collection:

- Space group - Please follow international table of Crystallography,
- Cell dimensions - Please check the use of points,
- $1/\sigma$: The high resolution shells in NleB2 and SseK1-UDP might be overestimated. I would suggest to process the data to high resolution and run few more cycles of refinement,
- Please add Wilson B-factor,

Refinement:

- In Resolution, please add the range for the high resolution shell,
- In number of reflections, please state if the number of reflections used in refinement and number of reflections for the high resolution shell,
- In R_{work}/R_{free} please add % unit and the corresponding values for the high resolution shell,
- Please add Number of reflections used for R_{free} , Ramachandran favoured (%), Ramachandran outliers (%) and Clashscore.

Reviewer #2 (Remarks to the Author):

The authors have done a great job attempting to refine the aMD by inclusion of GaMD functions. The work is well executed and manuscript is well structured and written. The findings of this report would definitely assist in the understanding of the arginine glycosylation protein structures and function. But there are minor issues with the paper in the present form.

#1. The abbreviation of "Gaussian Accelerated Molecular Dynamics" is "GaMD".

#2. In figure 4a, the authors compared the open and close forms of SeeK. How about the delta PMF between the open and close forms of SeeK ? The authors can use the GaMD results to calculate the delta PMF.

#3. In the method section, the authors should show the detail setting and information of GaMD.

Reviewer #3 (Remarks to the Author):

The manuscript describes structural and functional studies of an important class of bacterial glycosyltransferases that are used by these bacteria to specifically suppress host defense. This topic is of interest to a broad readership. The crystallographic studies are sound and reveal many details of donor-substrate enzyme interactions. Complementary biophysical methods, mainly NMR, have been used to gain more insight into the function of these interesting enzymes. The two most important conclusions of the studies are 1/ the retaining character of the enzymes can be experimentally proven, and 2/ the data support a front-face S_Ni type mechanism. MD simulations are finally used to present a comprehensive functional model.

The data presented are new and deserve publication. I also think that Nature Communications would be the right journal to present this study. However, there are some shortcomings and questions, which need to be addressed prior to publication as this is detailed in the following.

General comments:

I suggest that the authors phrase their statements on the proposed S_Ni mechanism more carefully, as no direct experimental evidence has been presented.

In general, ITC experiments provide enthalpic and entropic signatures of binding reactions. Unfortunately, this analysis has not been done although the raw data seem to be available. Some of the ITC thermograms in Supplementary Figure 1 look certainly like they are suitable for analysis of binding enthalpies and entropies. This full analysis should be done where possible. On the other hand, some of the thermograms, e.g. the ones reflecting binding of UDP-Glc and UPD-Gal seem so suffer from artifacts. What is the reason for the discontinuities and could these be corrected? Experimental details for the ITC experiments are missing and should be included. Data analysis with Origin also provides Chi² values for the dissociation constants, which should also be included. It would be interesting to compare the thermodynamic signatures of binding of activated sugars to these glycosyltransferases with existing literature data. If for some reason a full data analysis of the ITC data is impossible the authors should explain this.

Given that the importance of the experimental determination of the alpha-configuration of the D-GlcNAc-Arg linkage is so much stressed in the paper, why wasn't the complete set of chemical shifts of the glycopeptide obtained enzymatically reported? From the reading the paper I would assume this data set exists. Why not report it?

On p. 11, first para, it is mentioned that STD NMR experiments indicate a "significant conformational rearrangement of the peptide ligand". I do not agree with this conclusion. A change in binding epitopes may have many causes. Only one of them would be a conformational change.

On p. 11, second para, it is stated that "Hence, it is clear that differences in glycosylation of full length ...are not due to differences in binding modes ...". I simply cannot follow this argument, as glycosylated peptides or proteins have not been studied here.

On p. 15, Proposed mechanism: It is not quite correct that there is no experimental evidence for double displacement S_N2 . For human blood group B galactosyltransferase, a covalent intermediate has been described for the E303M mutant of this enzyme. There is a relatively recent review on this general topic by Ardevol et al. (2016). Maybe this is a good reference to get the non-expert reader involved.

Questions regarding Online Methods:

p. 20, NMR spectroscopy:

Which reference has been used for chemical shifts? For measurements in H₂O: how much D₂O has been added for the lock?

p.20, line 533: The strength of the B1 field and the flip angle used for the Gaussian pulse in the cascade should be given.

p.20, line 534: "spoil sequence" probably means spinlock filter. How long was the spinlock field?

p. 21, line 539: I find it a little surprising that at 2mM MnCl₂ good NMR spectra have been obtained. Is the concentration cited really OK? Compare the settings for the STD NMR experiments.

p. 21, line 538-547: What was the digital resolution of the 2D spectra used to determine the one-bond C-H coupling of C1 of GlcNAc?

Minor points:

p. 6, line 122: It should read "UDP-GlcNAc/MnCl₂"

p. 28, line 705: Here and throughout the text " α -GlcNAc" should be substituted with " α -D-GlcNAc"

Fig. 2: Replace "bounded" with "bound" or "bound to"; applies to Fig. 3, too.

p.6., line 130: "substrate" should be specified. It is the "donor substrate".

p. 8, line 182 to 185: The K_d for binding of UDP-GlcNAc to SseK2 wt vs. the F203A mutant was not reduced but increased. Sentence should be rephrased, e.g. "While the K_d measured for binding of UDP-GlcNAc to SseK1 F187A was almost the same ..., the K_d for binding of UDP-GlcNAc to ..."

p. 8, line 192: What is a "weaker K_d"?. Please correct the subscript "s".

p. 10, line 235: Substitute "bounded" with "bound".

p. 10, line 247: I could not find STD NMR spectra in the absence of Mn²⁺ and UDP. Either these spectra are included or it should be said "data not shown".

Supplementary Material:

The legends to Supplementary Figures 3 and 4 should include information on the field strength used (800 MHz), and the saturation time should be mentioned here. Applies to other legends of figures showing NMR data as well.

Supplementary Figure 16: In the table, please reduce to the number of significant digits.

Supplementary Figure 18: I do not understand the legend. What does it mean that an intermediate step has been shown by "chemical structure"?

Reviewer #4 (Remarks to the Author):

In this paper, Park et al reported the type III secretion system effector proteins of NleB family are retaining glycosyltransferases following a S_Ni mechanism. Recently, the structure of SseK3 was determined, revealing a GT-A fold. However, the specific enzyme mechanism and the identification of the catalytic base remain unclear. There are also discrepancies regarding whether these enzymes are retaining or inverting GTs because this has not been experimentally probed. In addition, details regarding substrate specificity based on structural evidence are also limited due to the lack of ternary complexes. Park et al want to address these remaining major questions of this field. However, there are several flaws need to be properly addressed. Validation the linkage form of the glycoside on the substrate protein is the key step to analyzing the enzymatic mechanism. The authors did not get the real structure of enzyme-sugar ligand-substrate ternary complex. And the substrate should be the full death domain proteins but not the peptides, because there is no evidence to show the peptides can be used as substrates and be hundred percent modified as the death domains (such as TRADD DD, FADD DD). So the methodology and the data cannot fully support the conclusion of the enzymatic mechanism of this newly reported post translational modification (GlcNAcylation on Arginine). I would like to recommend the editors of Nature communication considering a resubmission when the authors can provide crystal or NMR structures of the NleB/SseK-UDP-GlcNAc-DDs ternary complex.

Other points:

Q1: Line116. Based on protein sequence similarity, NleB belongs to the GT-8 family of enzymes, but one cannot predict the glycosyltransfer mechanism base on the protein sequence similarity since they have different modification residue, which means different niche around the modification site.

Q2: Line120. Utilize other method to verify the peptide modification site and percentage.

Q3: Line129. Show data of the protein quality. Do the mutations or truncation have defects on the enzyme activity of NleB family proteins?

Q4: If the authors use SseK1 to test the biochemistry activity and use SseK2 to do structural analyses, how to explain why sseK1 expressed in the cytoplasm whereas the SseK2 and SseK3 localized on Golgi apparatus?

Q5: Line175. Since the pi-pi stacking mode is not necessary for other GTs and slightly different in NleB/SseK123, is it required for the biochemistry and functional activities of NleB family? which pi-stacking is more or equally crucial ?

Q6: what is the correlation between the flexibility of lid domain and the enzyme activity?

Q7: what about binding affinity between peptides and the AxA mutant or other mutants? More experimental data are required to verify the molecular basis for peptide substrate recognition which the authors proposed.

Q8: SseK2 has the relatively weakest enzyme activity in all the reported family members, why the authors choose SseK2 to do the molecular docking? Can it represent the physiological situation? Only the open lid form of SseK2 can be used in docking, which means ternary complex and binary complex might have a lot of differences, especially in the enzyme-substrate binding aspect. If a short peptide cannot be docked in, we can speculate a full length death domain protein may induce much more conformational changes after ternary complex formation. So the docking model is not enough to clarify the enzymatic mechanism and enzyme-substrate coordination mechanism. The crystal structure of the ternary complexes are required.

Q9: Since WR motif is important for the binding of SseKs and peptides form death domains or GAPDH, loss of function data (in binding and modification) were largely lacking.

Q10: in Fig.6g, The E253A and HEN mutants showed different molecular weight pattern on the SDS PAGE, whereas in other gel (such as in Fig.6def) they did not show as this pattern, the authors should clarify it. The data quality of the immunoblotting in Fig6 should be improved.

Reviewers' comments:

Reviewer #1 (Remarks to the Author):

Park and co-workers report crystal structures of the bacterial effectors SseK1 and SseK2 from *Salmonella typhimurium* SL1344, and that of NleB2 from the *Escherichia coli* O145:H28, all involved in the glycosylation of arginine residues in host proteins. Specifically, the authors present (1) one crystal structure of SseK1 in complex with UDP at 3.64Å; (2) three crystal structures of SseK2: the unliganded form at 1.86Å, a binary complex with UDP at 1.66Å, and a binary complex with the sugar donor UDP-GlcNAc at 1.92Å; and (3) one crystal structure of the unliganded form of NleB2 at 2.3Å. SseK1, SseK2 and NleB2 display a typical GT-A fold. By using a combination of X-ray crystallography, STD-NMR, enzyme kinetics, molecular dynamics simulations, and in vivo experiments, the authors propose a plausible model for substrate/s binding and specificity, and catalytic mechanism. It is worth noting that the crystal structure of SseK3 was previously/very recently determined Esposito et al. *J. Biol. Chem.* 2018 Apr 6;293(14):5064-5078. However, the current work reported by Park and colleagues, is sufficiently novel and merits its publication in *Nature Communications*.

Some questions to be addressed and suggestions for improvement are listed below:

Q1. page 4, line 69: "GTs are classified according to their folding as GT-A, GT-B, or GT-C." A GT-D fold has been recently proposed. Please see:

Zhang H, Zhu F, Yang T, Ding L, Zhou M, Li J, Haslam SM, Dell A, Erlandsen H, Wu H. The highly conserved domain of unknown function 1792 has a distinct glycosyltransferase fold. *Nat Commun.* 2014 Jul 15;5:4339. Please could you comment on it?

Response: We added the information about the GT-D fold (Final ver: page 4, line 70/ Change-marked ver: line 70-71).

Q2. page 4, line 70: Most GT-A enzymes possess an DXD signature in which the carboxylates coordinate a divalent cation and/or a ribose. However, examples do now exist of enzymes from this fold family that do not possess the DXD signature. Please refer to

Lairson LL, Henrissat B, Davies GJ, Withers SG. Glycosyltransferases: structures, functions, and mechanisms. *Annu Rev Biochem.* 2008;77:521-55

Therefore, please modify the sentence accordingly.

Response: We rephrased the sentence (Final ver: page 4, line 71-72/ Change-marked ver: line 71-72).

Q3. page 4, line 71: The donor substrates include sugar-linked nucleotide diphosphates that also interact with the cofactor

Response: We rephrased the sentence (page 4, line 73-74/ Change-marked ver: line 73-74).

Q4. page 4, line 79: please define non-LEE.

Response: We defined 'non-LEE' (page 4, line 82-84/ Change-marked ver: line 86-89).

Q5. page 5, line 94: 'The T3SS effectors SseK1 and SseK2 from *Salmonella typhimurium* SL1344 are NleB orthologs that behave as NleB1-like GTs, although they differ in protein substrate specificity'. I would suggest the authors to introduce this information in the abstract. In addition, 'specificity' by 'specificity'.

Response: We introduced this information in the abstract (page 3, line 52-54/ Change-marked ver: line 52-54) and corrected the spelling mistake.

Q6. page 6, line 124: Please replace by 'We therefore provide compelling experimental evidence by NMR that

the transfer of GlcNAc by SseK1 follows a retaining mechanism.' In addition, explain why you consider that the SseK2 is a retaining GT.

Response: We rephrased the paragraph (line 126-129/ Change-marked ver: line 136-138). We have clearly demonstrated by NMR that the configuration of the sugar is retained after the reaction, so these enzymes follow a retaining mechanism. In addition (see Supplementary Figure 15) all the catalytic machinery is conserved along this family of enzymes. For example, the conserved His and Glu residues forming the HEN motif likely interact with the acceptor arginine and hence it is plausible that all these orthologs follow a retaining mechanism, just like SseK1

Q7. page 6, line 128: Please replace the title 'GT overall architecture' by 'Overall architecture of SseK1, SseK2 and NleB2 GTs'

Response: We replaced it as suggested. (page 6, line 131/ Change-marked ver: line 140)

Q8. page 6, line 130 and 136: Instead, please state the form in which the crystal structure was obtained for each protein. For example: NleB2 was solved in its unliganded form. SseK1 was solved in complex with UDP. SseK2 was solved in its unliganded form and in complex with UDP and UDP-GlcNAc.

Response: We specified all of the forms in which crystals were obtained. (page 6, line 132-134/ Change-marked ver: line 141-145)

Q9. page 6, line 132: 'the N-terminus of SseK1 and SseK2 were truncated, respectively.' Please explain the reason of that.

Response: We explained the reason for truncating the proteins for crystallography. (page 6, line 135-138/ Change-marked ver: line 146-149)

Q10. page 7, line 143: 'because their datasets were obtained at a higher resolution.' And the authors have three different snapshots of the active site.

Response: We rephrased the paragraph. (page 7, line 149-150/ Change-marked ver: line 168-170)

Q11. page 7, line 145: 'root-mean-square deviation value of the structure is ~2.0 Å based on DALI pairwise comparison'. Please include the r.m.s.d. values, Z-score as well as number of compared residues.

Response: We added the numbers as suggested. (page 7, line 152-153/ Change-marked ver: line 172-173)

Q12. page 7, line 146: 'Most GT-A fold GTs are single domain proteins that contain a Rossmann-like fold.', 'which has two abutting $\beta/\alpha/\beta$ Rossmann-like domains ($\beta_3-\alpha_2-\beta_4-\alpha_3-\beta_5$)', 'SseK2 can be divided into three types of sub-domains, namely'. Please clarify. I would suggest to include residue numbers to support the 'domain' and/or subdomains description.

Response: We rephrased the paragraph. (page 7, line 157-159/ Change-marked ver: line 177-179)

Q13. page 7, line 156: 'However, in the complex structure with the UDP or UDP-GlcNAc, the substrate leads to an unambiguous electron density map for the C-terminal lid domain.'. Please specify the protein complexes.

Response: We specified the complexes and corrected the sentence. (page 7, line 164/ Change-marked ver: line 183-184)

Additional response: We corrected the sentence from 'UDP or UDP-GlcNAc' to 'UDP and UDP-GlcNAc'. From this, we want to explain that the electron density map of C-terminal lid domain was shown both in UDP bound structure and UDP-GlcNAc bound structure.

Q14. page 8, line 186: "In the SseK3 structure, SseK2-like π - π stacking interaction is conserved and Phe190SseK3 and Trp52SseK3 (corresponding to Phe203SseK2 and Trp65SseK2) participate in an interaction with the uridine group in the same orientation." A good opportunity to introduce a Figure 3 panel showing the arrangement of these aromatic residues into the SseK3 active site.

Response: We changed Figure 3 as suggested.

We added a panel of uridine binding mode of SseK3 in figure 3a and please check the upper picture. The red rectangle is the part we added. In addition, we described the related content in manuscript line 195-198 (Change-marked ver: line 220-221).

Q15. page 9, line 214: What is the experimental evidence that support the substrate binding/product release order for SseK1, SseK2, SseK3 or NleB2? I agree with the authors that in most of the cases, GTs follow an order mechanism with the donor substrate binding first. However, in the absence of experimental enzymatic/kinetic data, the authors should be cautious with the interpretation of the structural data. A statement should be added accordingly.

Response: We agree with the reviewer that we do not have any experimental evidence on whether these enzymes follow an ordered mechanism. However, we do not mention in the text anything related to an ordered mechanism. We only state that in the presence of the sugar nucleotide, we see large conformational changes of the C-terminal lid domain. This also occurs in similar virulence factors such as the *Legionella pneumophila* glucosyltransferase and the toxin A and B (PMIDs: 20030628 and 17901056), which share this C-terminal lid.

Q16. page 9, line 229: 'Based on these data, we suggest that the conformation of the C-terminal domain is highly similar between SseK2 and SseK3.' Please remove this sentence.

Response: We deleted that sentence.

Q17. page 10, line 245: 'STD NMR experiments of SseK1 and SseK2 in the presence of the peptides FADD110-118, TRADD229-237, and GAPDH195-203 showed that all three peptides bound to both enzymes in the presence and absence of Mn²⁺ and UDP'. Please rephrase the sentence.

Response: We have rephrased the sentence (page 10-11, lines 263-268/ Change-marked ver: line 288-295) "Standard homo- and heteronuclear 2D NMR techniques were used to obtain the chemical shift assignments of GAPDH195-203, FADD110-118, and TRADD229-237 (Supplementary Tables. 3-5). For

each peptide, four different enzyme systems were prepared: apo-SseK1, apo-SseK2, holo-SseK1, and holo-SseK2, where apo and holo stand for the enzyme without and with Mn²⁺ and UDP, respectively. We observed that all three peptides bound to both SseK1 and SseK2, irrespectively of the forms used in the experiments (Supplementary Figs. 3, 4). These data imply that binding of the short peptide ligands occurs independently of enzymatic activity and can also take place in the absence of the sugar nucleotide". The text is more elaborated than before and it is clear that the peptides can bind independently of the presence of the nucleotide and cofactor. As you can see we have elaborated more what was stated before. We think that the sentence is more understandable now and can be unambiguously interpreted.

Q18. page 10, line 254: 'notion that that a' by 'notion that a'

Response: We corrected this sentence. (page 11, line 275/ Change-marked ver: line 302)

Q19. page 13, line 323: I would suggest the authors to introduce here a section focused into the 'Catalytic mechanism of SseK1, SseK2 and NleB2'. In that context, please move/introduce the entire 'Anomeric configuration of glycosylated peptides' section here, and renumber Figure 1 accordingly. After the discussing the catalytic mechanism, the authors could introduce the section 'Catalytic importance of HEN motif'.

Response: We thank the reviewer for this suggestion. However, we believe that the current order of our manuscript is easier to read and we prefer to discuss the HEN motif before the mechanism because we can then explain why we think an S_Ni mechanism is more plausible than an S_N2 mechanism. The results for HEN motif mutants rule out completely the possibility of the S_N2 mechanism and support the S_Ni mechanism, together with the molecular dynamics simulations.

Q20. page 15, line 394. 'Two reaction mechanisms, SN2 and SNi, have been proposed for retaining GTs.' Please see:

PLoS One. 2013 Aug 1;8(8):e71077. doi: 10.1371/journal.pone.0071077. Print 2013. Geometric attributes of retaining glycosyltransferase enzymes favor an orthogonal mechanism. Schuman B, Evans SV, Fyles TM.

Please, could you comment on this?

Response: We agree with the reviewer and we have changed this accordingly in the main manuscript. Now, we state that three reactions mechanisms, S_N2, S_Ni, and orthogonal, have been proposed for retaining GTs. However, there are no experimental evidences for the S_N2 and the orthogonal mechanism. This has been also discussed in the main text. (page 16-17, line 441-449/ Change-marked ver: line 468-478)

Q21. page 17, line 432: 'these residues are important in catalysis' by "these residues impact enzyme catalysis".

Response: We corrected this sentence. (page 18, line 484/ Change-marked ver: line 531)

Q22. page 18, line 470: I would remove the last paragraph 'In summary...'.

Response: We removed this paragraph as suggested.

Q23. page 19, line 480: *

Response: We added the amino acid sequences in 'Supplementary Table 2' as suggested.

Q24. Supplementary Table 1.

Data Collection:

Space group - Please follow international table of Crystallography,

Response: We corrected this table.

- Cell dimensions - Please check the use of points,

- I/signal: The high resolution shells in NleB2 and SseK1-UDP might be overestimated. I would suggest to process the data to high resolution and run few more cycles of refinement,

- Please add Wilson B-factor,

Response: We corrected and added this information as suggested.

Q25.Refinement:

- In Resolution, please add the range for the high resolution shell,

- In number of reflections, please state if the number of reflections used in refinement and number of reflections for the high resolution shell,

- In Rwork/Rfree please add % unit and the corresponding values for the high resolution shell,

- Please add Number of reflections used for Rfree, Ramachandran favoured (%), Ramachandran outliers (%) and Clashscore.

Response: We corrected and added this information as suggested.

Reviewer #2 (Remarks to the Author):

The authors have done a great job attempting to refine the aMD by inclusion of GaMD functions. The work is well executed and manuscript is well structured and written. The findings of this report would definitely assist in the understanding of the arginine glycosylation protein structures and function. But there are minor issues with the paper in the present form.

#1. The abbreviation of “Gaussian Accelerated Molecular Dynamics” is “GaMD”.

Response: We corrected the abbreviation.

#2. In figure 4a, the authors compared the open and close forms of SseK. How about the delta PMF between the open and close forms of SseK? The authors can use the GaMD results to calculate the delta PMF.

Response: The GaMD simulations of SseK1 and SseK2 and the SseK2/FADD complex were performed on the closed forms of the enzymes. During the simulations of the free enzymes, SseK1 and SseK2, which were performed over 800 ns each, we did not observe any transition from the closed to open conformations. Therefore, we cannot calculate the delta PMF of this transition.

#3. In the method section, the authors should show the detail setting and information of GaMD.

Response: We have included all the relevant details about the setting of the GaMD simulations (page 24, lines 657-663/ Change-marked ver: line 698-704)

Reviewer #3 (Remarks to the Author):

The manuscript describes structural and functional studies of an important class of bacterial glycosyltransferases that are used by these bacteria to specifically suppress host defense. This topic is of interest to a broad readership. The crystallographic studies are sound and reveal many details of donor-substrate enzyme interactions. Complementary biophysical methods, mainly NMR, have been used to gain more insight into the function of these interesting enzymes. The two most important conclusions of the studies are 1/ the retaining character of the enzymes can be experimentally proven, and 2/ the data support a front-face S_Ni type mechanism. MD simulations are finally used to present a comprehensive functional model. The data presented are new and deserve publication. I also think that Nature Communications would be the right journal to present this study. However, there are some shortcomings and questions, which need to be addressed prior to publication as this is detailed in the following.

General comments:

I suggest that the authors phrase their statements on the proposed S_Ni mechanism more carefully, as no direct experimental evidence has been presented.

Response: We have revised this in the updated version of the manuscript. Our data conflicts with an S_N2 mechanism because there are no residues that could act as nucleophiles in the vicinity of the GlcNAc moiety. The neighboring residues forming the HEN motif are not essential for catalysis, implying that these enzymes do not follow an S_N2 mechanism. In addition, we also discuss that a similar mechanism to the S_Ni was proposed earlier that also involves a front face mechanism. However, this mechanism has never been probed experimentally. Thus, we believe that these enzymes might follow the most accepted and typical S_Ni mechanism for the retaining glycosyltransferases. In any case, we have toned down our claims regarding the proposed S_Ni mechanism.

We think it is clear that we have toned down our claims with respect to the mechanism because we now suggest that they might follow the S_Ni mechanism. We think that this is now well explained in the revised version of our manuscript (especially line 441-470/ Change-marked ver: line 468-478).

In general, ITC experiments provide enthalpic and entropic signatures of binding reactions. Unfortunately, this analysis has not been done although the raw data seem to be available. Some of the ITC thermograms in Supplementary Figure 1 look certainly like they are suitable for analysis of binding enthalpies and entropies. This full analysis should be done where possible. On the other hand, some of the thermograms, e.g. the ones reflecting binding of UDP-Glc and UDP-Gal seem so suffer from artifacts. What is the reason for the discontinuities and could these be corrected? Experimental details for the ITC experiments are missing and should be included. Data analysis with Origin also provides χ^2 values for the dissociation constants, which should also be included. It would be interesting to compare the thermodynamic signatures of binding of activated sugars to these glycosyltransferases with existing literature data. If for some reason a full data analysis of the ITC data is impossible the authors should explain this.

Response: We thank the reviewer for this suggestion. We added extra data that include the stoichiometry, enthalpy, entropy, Gibbs free energy, and binding affinity in 'Supplementary Figure 1' as suggested. We purified the enzyme again and performed ITC assays with UDP-Glc and UDP-Gal. These data support our earlier analysis and lack artifacts.

Given that the importance of the experimental determination of the alpha-configuration of the D-GlcNAc-Arg linkage is so much stressed in the paper, why wasn't the complete set of chemical shifts of the glycopeptide obtained enzymatically reported? From the reading the paper I would assume this data set exists. Why not report it?

Response: Chemical shift assignments were made for the 9-mer peptides (FADD₁₁₀₋₁₁₈, TRADD₂₂₉₋₂₃₇, and GAPDH₁₉₅₋₂₀₃) used for STD NMR. We have now updated the manuscript to include these assignments in the supplementary material (Supplementary Tables 3-5). However, assignment of the glycopeptide was not possible since a very low amount of the glycosylated 16-mer of GAPDH was

obtained from the enzymatic reaction for the NMR study, as the reaction did not proceed quantitatively, which meant that the final sample consisted of a complex mixture which additionally contained the non-glycosylated peptide, a large excess of free UDP-GlcNAc, and the hydrolysis products UDP and alpha- and beta-GlcNAc (as shown in Fig.1-left in the main text), making extremely difficult to carry out a full chemical shift assignment of the glycopeptide. Instead, we were able to confirm the linkage between GlcNAc and arginine in the resulting glycopeptide through a 2D $^1\text{H}, ^1\text{H}$ -TOCSY experiment which showed a correlation between the GlcNAc anomeric proton and a η -proton of an arginine. Given that the glycosylation has been confirmed through other biophysical techniques within this paper (i.e. kinetics experiments with GAPDH wt and mutants), we believe that this is sufficient evidence to show that the measured coupling indeed belongs to the GlcNAc that is covalently linked in alpha configuration to an arginine residue of the peptide ligand. We have now added to figure 1 an expansion of the TOCSY experiment showing the above mentioned correlation between the anomeric proton and the arginine proton.

Q1. On p. 11, first para, it is mentioned that STD NMR experiments indicate a "significant conformational rearrangement of the peptide ligand". I do not agree with this conclusion. A change in binding epitopes may have many causes. Only one of them would be a conformational change.

Response: We agree with the reviewer that, in general, a change in epitope is not sufficient to claim a conformational change, since the differences may be due to a different binding mode. However, in this case, we believe that a change in conformation can be claimed, since the WR-motif is preserved as the predominant contact, ruling out the possibility of a significant change in binding mode. This is further evidenced by the fact that there is no substantial change in absolute STD intensities.

Q2. On p. 11, second para, it is stated that "Hence, it is clear that differences in glycosylation of full length ...are not due to differences in binding modes ...". I simply cannot follow this argument, as glycosylated peptides or proteins have not been studied here.

Response: Based on the similarities of how the peptides interacted with the enzymes from STD experiments and the differences in substrate glycosylation, we suggest that the differences in activity for the SseK1 and SseK2 for the protein substrates are due to contacts with the substrates far away from the active binding site. Although we do not have any direct experimental evidence for this, our molecular dynamics simulations for SseK2 in complex with FADD death domain suggest that FADD residues interact with regions of the SseK2 far distant from the active site. In any case, we have toned down our argument in the revised version of our manuscript.

We now suggest that our claims are more likely to happen because as we explain we only have indirect evidences of it. These are our own STD-NMR experiments and molecular dynamics simulations. As well as with the $\text{S}_{\text{N}}\text{i}$ mechanism, we have toned down our claims because we now suggest that interactions far away from the active site between SseK1/2 and FADD/TRADD/GAPDH are likely to happen explaining the differences among these enzymes regarding the glycosylation of protein substrates.

Q3. On p. 15, Proposed mechanism: It is not quite correct that there is no experimental evidence for double displacement $\text{S}_{\text{N}}2$. For human blood group B galactosyltransferase, a covalent intermediate has been described for the E303M mutant of this enzyme. There is a relatively recent review on this general topic by Ardevol et al. (2016). Maybe this is a good reference to get the non-expert reader involved.

Response: The reviewer is partially right. That study claimed they demonstrated a $\text{S}_{\text{N}}2$ mechanism for the human blood group B galactosyltransferase. However, more robust experimental work has clearly demonstrated that the $\text{S}_{\text{N}}2$ mechanism was not right. In fact, the results with E303M appear to be an artifact because the later study (see PMID:28960760 or DOI:[10.1002/anie.201707922](https://doi.org/10.1002/anie.201707922)) clearly demonstrates that this enzyme follows the typical $\text{S}_{\text{N}}\text{i}$ mechanism. In fact, they suggest that most retaining glycosyltransferases must follow the $\text{S}_{\text{N}}\text{i}$ mechanism. The authors claim that E303 must play an important role in the stabilization of the transition state and product release.

Questions regarding Online Methods:

Q4.p. 20, NMR spectroscopy: Which reference has been used for chemical shifts? For measurements in H₂O: how much D₂O has been added for the lock?

Response: the residual HDO signal was used as a reference. For the measurements made in H₂O, the ratio was 90% H₂O/10% D₂O. The methods section has been modified accordingly (page 21-22, lines 576-582/ Change-marked ver: line 614-627)

Q5.p.20, line 533: The strength of the B₁ field and the flip angle used for the Gaussian pulse in the cascade should be given.

Response: A power of 0.4 mW was used for the Gaussian shaped pulse constituting the repetitive element of the train of selective pulses for saturation, corresponding to a B₁ field strength of 78 Hz. For a shaped pulse length of 50 ms leads to a flip angle of 1,150° (12.8 times the 90° pulse). The methods section has been modified accordingly (page 22, lines 583-589/ Change-marked ver: line 614-627).

Q6.p.20, line 534: "spoil sequence" probably means spinlock filter. How long was the spinlock field?

Response: The spoil sequence described here is a sequence of two trim pulses of 2.5 and 5 ms followed by a 40% z-gradient applied for 3 ms at the beginning of the experiment to destroy any residual x,y-magnetisation from the previous scan. A spinlock filter was also used to filter out protein signals and was applied for 40 ms. The methods section has been modified accordingly (lines 585-588/ Change-marked ver: line 614-627).

Q7.p. 21, line 539: I find it a little surprising that at 2mM MnCl₂ good NMR spectra have been obtained. Is the concentration cited really OK? Compare the settings for the STD NMR experiments.

Response: We agree with the referee that this issue was a bit confusing in the way it was written in the methods section. This concentration of MnCl₂ is referring to the experimental conditions for the preparation of the glycosylated peptide, not for the NMR experiments. The glycopeptide was purified previous to the NMR study. Nevertheless, some remaining amount of Mn²⁺ was evident as broadening of the NMR signals, but not to such an extent as to preclude observation of cross peaks in the 2D NMR spectra.

We have modified the starting sentence at line 590 (Change-marked ver: line 630) to "Samples for peptide glycosylation were prepared...", as well as the caption of Figure 1.

Q8.p. 21, line 538-547: What was the digital resolution of the 2D spectra used to determine the one-bond C-H coupling of C1 o GlcNAc?

Response: The original experiment at 800 MHz had a resolution in the direct dimension of 20 Hz, which, although still qualitatively valid to distinguish the coupling, was not of enough resolution as to quantitatively report the coupling. We have now repeated the experiment at 500 MHz with a resolution of 1.6 Hz and measured the coupling to be 169 Hz. The manuscript has been updated to include the spectrum acquired at 500 MHz (Figure 1, and lines 595-599/ Change-marked ver: line 633-638)

Minor points:

Q9.p. 6, line 122: It should read "UDP-GlcNAc/MnCl₂"

Response: We corrected this.

Q10.p. 28, line 705: Here and throughout the text "α-GlcNAc" should be substituted with "α-DGlcNAc"

Response: We corrected this.

Q11.Fig. 2: Replace "bounded" with "bound" or "bound to"; applies to Fig. 3, too.

Response: We corrected this.

Q12.p.6., line 130: "substrate" should be specified. It is the "donor substrate".

Response: We corrected this.

Q13.p. 8, line 182 to 185: The K_d for binding of UDP-GlcNAc to SseK2 wt vs. the F203A mutant was not reduced but increased. Sentence should be rephrased, e.g. "While the K_d measured for binding of UDP-GlcNAc to SseK1 F187A was almost the same ..., the K_d for binding of UDP-GlcNAc to ..."

Response: We rephrased this paragraph as suggested.

Q14.p. 8, line 192: What is a "weaker K_d "?. Please correct the subscript "s".

Response: We corrected this sentence.

Q15.p. 10, line 235: Substitute "bounded" with "bound".

Response: We corrected this.

Q16.p. 10, line 247: I could not find STD NMR spectra in the absence of Mn^{2+} and UDP. Either these spectra are included or it should be said "data not shown".

Response: Those spectra were indeed in Fig 5c and Supplementary Fig 6, but these figures have now been relabelled. There was no Mn^{2+} or UDP in these samples, though they were labelled as having Mn^{2+} . We apologise and thank the reviewer for noticing our mistake. Figure 5 and Supplementary Figure 6 have been modified accordingly

Supplementary Material:

Q17.The legends to Supplementary Figures 3 and 4 should include information on the field strength used (800 MHz), and the saturation time should be mentioned here. Applies to other legends of figures showing NMR data as well.

Response: We have modified the legends to include that information.

Q18.Supplementary Figure 16: In the table, please reduce to the number of significant digits.

Response: We corrected this.

Q19.Supplementary Figure 18: I do not understand the legend. What does it mean that an intermediate step has been shown by "chemical structure"?

Response: We corrected this.

Reviewer #4 (Remarks to the Author):

In this paper, Park et al reported the type III secretion system effector proteins of NleB family are retaining glycosyltransferases following a S_Ni mechanism. Recently, the structure of SseK3 was determined, revealing a GT-A fold. However, the specific enzyme mechanism and the identification of the catalytic base remain unclear. There are also discrepancies regarding whether these enzymes are retaining or inverting GTs because this has not been experimentally probed. In addition, details regarding substrate specificity based on structural evidence are also limited due to the lack of ternary complexes. Park et al want to address these remaining major questions of this field. However, there are several flaws need to be properly addressed. Validation the linkage form of the glycoside on the substrate protein is the key step to analyzing the enzymatic mechanism. The authors did not get the real structure of enzyme-sugar ligand-substrate ternary complex. And the substrate should be

the full death domain proteins but not the peptides, because there is no evidence to show the peptides can be used as substrates and be hundred percent modified as the death domains (such as TRADD DD, FADD DD). So the methodology and the data cannot fully support the conclusion of the enzymatic mechanism of this newly reported post translational modification (GlcNAcylation on Arginine). I would like to recommend the editors of Nature communication considering a resubmission when the authors can provide crystal or NMR structures of the NleB/SseK-UDP-GlcNAc-DDs ternary complex.

Other points:

Q1: Line116. Based on protein sequence similarity, NleB belongs to the GT-8 family of enzymes, but one cannot predict the glycosyltransfer mechanism base on the protein sequence similarity since they have different modification residue, which means different niche around the modification site.

Response: We have deleted this statement.

Q2: Line120. Utilize other method to verify the peptide modification site and percentage.

Response: In addition to the determination of the configuration of the anomeric carbon of the transferred GlcNAc, by undecoupled HSQC experiments, we have now been able to confirm that the linkage takes place between GlcNAc and arginine in the resulting glycopeptide through a 2D ¹H,¹H-TOCSY experiment which showed a correlation between the GlcNAc anomeric proton and a η-proton of an arginine. An expansion of this spectrum is now included in Figure 1.

Q3: Line129. Show data of the protein quality. Do the mutations or truncation have defects on the enzyme activity of NleB family proteins?

Response: As we described in manuscript, we truncated the flexible N-termini and mutated a Cys residue to make protein crystals in obtain the 3D structure. However, for other In-vivo and In-vitro assays, we have used the wild-type form of the proteins. In addition, the mutated Cys residue is located on the opposite side from the active site, implying that this Cys is a scaffold residue and is not involved in catalysis.

Q4: If the authors use SseK1 to test the biochemistry activity and use SseK2 to do structural analyses, how to explain why sseK1 expressed in the cytoplasm whereas the SseK2 and SseK3 localized on Golgi apparatus?

Response: We did not make such a statement nor did we assay for the subcellular localization of these enzymes. Differences in subcellular localization reported by others may be due to protein-protein interactions other than those that are the subject of investigation here.

Q5: Line175. Since the pi-pi stacking mode is not necessary for other GTs and slightly different in NleB/SseK123, is it required for the biochemistry and functional activities of NleB family? which pi-stacking is more or equally crucial ?

Response: We thank the reviewer for this suggestion. We have added ITC data and NF- κ B assay result in Supplementary Figure 1 and have updated this in the revised manuscript. Based on our data, the π - π stacking mode is important for binding to UDP-GlcNAc and enzyme activity (Supplementary Figure 1b, c and d). In addition, combined with the ITC assay and the NF- κ B activation assay, Trp65^{SseK2} and Trp51^{SseK1} play a more crucial role than Phe203^{SseK2} and Trp331^{SseK1} to UDP-GlcNAc binding and enzyme activity. (page 8-9, line 198-205/ Change-marked ver: line 220-228)

Q6: what is the co-relation between the flexibility of lid domain and the enzyme activity?

Response: We thank the reviewer for this question. We revealed the crystal structure of SseK2 with and without UDP-GlcNAc and as a result, the open and closed conformation were found out. This substrate-induced conformational change in glycosyltransferase has been well studied (PMID: 15653326). Similar structures in which the structural change of the C-terminal lid-domain has been already published (PMID: 11419947, 12051854, 11032794 and 11592969) and for all of these enzymes, the conformational change in the flexible loop generates a helical structure in the C-terminal region similar to SseK2. Through the lid domain regulation, the sugar donor is stabilized and buried deep in the catalytic pocket and blocks the water molecule entry to ensure the precise chemical reaction takes place. To demonstrate the function of the lid domain, we added additional ITC data in Supplementary Figure 1b, c which show that the lid domain truncated form of SseK1 increased the K_d value to UDP-GlcNAc as compared to the wild type for about 155-fold. This result shows that the C-terminal lid domain functions to facilitate substrate binding and this correlates with the structural data (Figure 4). (page 10, line 243-246/ Change-marked ver: line 266-269)

Q7: what about binding affinity between peptides and the AxA mutant or other mutants? More experimental data are required to verify the molecular basis for peptide substrate recognition which the authors proposed.

Response: Similar experiments in which the DxD was mutated to AxA mutant have been already performed for other virulence factors such as the *Legionella pneumophila* glycosyltransferase and demonstrated that there is a 10-fold decrease in binding to the sugar nucleotide and no activity towards protein substrates (PMID: 20030628). Hence, we do not think that these experiments are required to confirm the importance of the DxD in binding and catalysis. In addition, numerous examples of the DxD motif importance in binding and catalysis exist in the literature for other GT-A glycosyltransferases. We previously demonstrated the importance of the DxD motif in catalysis (PMID: 23332158). Regarding other residues, we also demonstrate by kinetic experiments the importance of the HEN motif in binding and catalysis towards the substrates. Finally, our STD-NMR experiments and the importance of the WR motif in the peptides are supported by molecular dynamics simulations. To fully understand the molecular basis of the protein substrate recognition, we would need the crystal structure of a ternary complex and this is something that we have not been able to obtain, despite 18 months of effort (please see below).

Q8: SseK2 has the relatively weakest enzyme activity in all the reported family members, why the authors choose SseK2 to do the molecular docking? Can it represent the physiological situation? Only the open lid form of SseK2 can be used in docking, which means ternary complex and binary complex might have a lot of differences, especially in the enzyme-substrate binding aspect. If a short peptide cannot be docked in, we can speculate a full length death domain protein may induce much more conformational changes after ternary complex formation. So the docking model is not enough to clarify the enzymatic mechanism and enzyme-substrate coordination mechanism. The crystal structure of the ternary complexes are required.

Response: SseK1 preferentially glycosylates GAPDH and TRADD while SseK2 glycosylates FADD. The targeted region of GAPDH is a random coil, so it is difficult to model. Conversely, the targeted regions of TRADD and FADD are helical, making modeling feasible. This is why we chose SseK2 for molecular docking studies. FADD was chosen because a high-resolution crystal structure exists. Regarding the likelihood of conformational changes upon formation of the ternary complex, our 500 ns GaMD simulation of the SseK2/ FADD (full protein) complex, showed theoretically that at that timescale there are no significant conformational rearrangements.

We agree with the reviewer that the crystal structure of the ternary complexes would clarify the issue about conformational changes upon complex formation. However, we have been attempting for nearly 18 months to obtain this complex and have exhausted all known combinations of protein truncations, point mutants, and buffer conditions. We have attempted 1) co-crystallisation experiments of SseK1/SseK2 with UDP-MnCl₂ and different peptides, 2) soaking experiments of peptides on crystals of SseK1-UDP-Mn⁺² or SseK2-UDP-Mn⁺², 3) cocrystallization experiments of SseK1/SseK2 with UDP and the FADD death domain, 4) two fusion proteins containing SseK1/2 coupled to the death domain of FADD, and 5) several SseK1/2 fusion proteins coupled to the peptide substrates.

Finally, we have attempted similar experiments for other GT-A fold glycosyltransferases. In particular, we worked with the *Legionella pneumophila* glucosyltransferase (binary complexes were published in PMID: 20030628) and the toxin A and B glycosyltransferases, and in both cases we were never able to obtain ternary complexes.

Q9: Since WR motif is important for the binding of SseKs and peptides from death domains or GAPDH, loss of function data (in binding and modification) were largely lacking.

Response: We thank the reviewer for this question. To reveal the importance of the GAPDH WR motif, we did additional experiment for measuring the enzyme kinetics by using GAPDH peptides including the wild type and WR mutant peptides. As a result, the single mutation form of each WR motif decreases the catalytic efficacy (40% and 47% to Trp and Arg, respectively) as compared to the wild type and the double mutant catalytic efficacy synergistically decreased (17%) (Supplementary Figure 16b). This data supports the STD-NMR experiment and combined with the kinetics assay and STD-NMR results, it can be concluded that the WR motif is important for the interaction with SseK. Additionally, we dissected the contribution of the WR-motif to binding affinity, by running STD NMR competition experiments between WT and single/double mutants of the TRADD peptide interacting with SseK1. The relevance of the WR-motif was demonstrated as any modification on the WR-motif impacted negatively the affinity of the peptide for the enzyme (new text included; lines 287-310 (Change-marked ver: line 313-336); Supplementary Figures 19 and 20)

Q10: in Fig.6g, The E253A and HEN mutants showed different molecular weight pattern on the SDS PAGE, whereas in other gel (such as in Fig.6def) they did not show as this pattern, the authors should clarify it. The data quality of the immunoblotting in Fig6 should be improved.

Response: While not the subject of this work, we and others (e.g. Esposito, JBC, 2018), have observed that some, but not all of the NleB/SseK orthologs are self-glycosylated. We believe that self-glycosylation may contribute to the differential migration of some proteins in SDS-PAGE. We have carefully checked our immunoblotting data for rigor and reproducibility and believe that the images provided are of high quality and support the overall conclusions reached in the manuscript.

Further modifications:

- Figure 5, Supp. Fig. 5, Supp. Fig. 6:
 - o In FADD peptide Val116 was wrongly labelled: it has been corrected to Ala116
 - o In TRADD peptide Ala233 was wrongly labelled: it has been corrected to Val233
 - o The STD NMR experiments for TRADD have been repeated with the correct sequence, ending in Leucine (KWRKVG**R**S**L**). Previously, the sequence ending with Isoleucine (KWRKVG**R**S**I**) was wrong.

REVIEWERS' COMMENTS:

Reviewer #1 (Remarks to the Author):

The authors have been answer my concerns satisfactorily.

Reviewer #3 (Remarks to the Author):

The questions of the reviewers have been answered adequately and new experimental data have been added. I think, the authors have done a great job in improving the manuscript, and I recommend publication of the revised manuscript without further changes.